# An analysis of Ermakov-Zolotukhin quadrature using kernels

**Ayoub Belhadji**
Univ Lyon, ENS de Lyon
Inria, CNRS, UCBL
LIP UMR 5668, Lyon, France
`ayoub.belhadji@ens-lyon.fr`

## Abstract

We study a quadrature, proposed by Ermakov and Zolotukhin in the sixties, through the lens of kernel methods. The nodes of this quadrature rule follow the distribution of a determinantal point process, while the weights are defined through a linear system, similarly to the optimal kernel quadrature. In this work, we show how these two classes of quadrature are related, and we prove a tractable formula of the expected value of the squared worst-case integration error on the unit ball of an RKHS of the former quadrature. In particular, this formula involves the eigenvalues of the corresponding kernel and leads to improving on the existing theoretical guarantees of the optimal kernel quadrature with determinantal point processes.

## 1 Introduction

Integrals appear in many scientific fields as quantities of interest per se. For example, in statistics, they represent expectations [27], while in mathematical finance, they represent the prices of financial products [17]. Unfortunately, integrals that can be written in closed form are exceptional. In general, their values are only known through approximations. For this reason, numerical integration is at the heart of many tasks in applied mathematics and statistics. Among all the possible approximation schemes, quadratures are the most practical since they approximate the integral of a function by a finite mixture of its evaluations. In this work, we focus on quadrature rules that take the form

$$\int_{\mathcal{X}} f(x)g(x)\mathrm{d}\omega(x) \approx \sum_{i \in [N]} w_i f(x_i), \tag{1}$$

where the nodes $x_i$ are independent of $f$ and $g$, while the weights $w_i$ depend only on $g$. The nodes and the weights of a quadrature may be seen as degrees of freedom that the practitioner may tune in order to achieve a given level of approximation error. The design of quadratures gave birth to a rich literature from Gaussian quadrature [15] to Monte Carlo methods [25] to quadratures based on determinantal point processes (DPPs) [1]. These latter form a large class of probabilistic models of repulsive random subsets that make numerical integration possible in a variety of domains with strong theoretical guarantees. In particular, central limit theorems with asymptotic convergence rates that scale better than the typical Monte Carlo rate $\mathcal{O}(N^{-1/2})$ were proven for several DPP based quadratures: when the integrand is a $\mathcal{C}^1$ function [1] or even when the integrand is non-differentiable [10]. Moreover, it is possible to design quadrature rules based on DPPs with non-asymptotic guarantees and with rates of convergence that adapt to the smoothness of the integrand. This is the case of the quadrature proposed by Ermakov and Zolotukhin in [14] and recently revisited in [16], and the optimal kernel quadrature [3].

In this work, we study the quadrature rule proposed by Ermakov and Zolotukhin (EZQ) through the lens of kernel methods. We start by comparing the weights of EZQ to the weights of the optimal

kernel quadrature (OKQ), and we prove that they both belong to a broader class of quadrature rules that we call *kernel based interpolation quadrature*. Then, we study the approximation quality of EZQ in reproducing kernel Hilbert spaces (RKHSs). This is done by proving a general tractable formula of the expected value of the squared worst-case integration error for functions that belong to the unit ball of an RKHS when the nodes follow the distribution of a determinantal point process. This formula involves principally the eigenvalues of the integral operator, and converges to 0 at a slightly slower rate than the optimal rate. Interestingly, this analysis yields a better upper bound for the optimal kernel quadrature with DPPs proposed initially in [3]. Comparably to the theoretical guarantees given in [14, 16], our theoretical guarantees are independent of the choice of the test function. This facilitates the comparison of EZQ with other quadratures such as OKQ.

The rest of the article is organized as follows. Section 2 reviews the work of [14] and recall key concepts on kernel based quadrature. In Section 3, we present the main results of this work and their consequences. A sketch of the proof of the main theorem is given in Section 4. We illustrate the theoretical results by numerical experiments in Section 5. Finally, we give a conclusion in Section 6.

**Notation and assumptions.**   We use the notation $\mathbb{N}^* = \mathbb{N} \smallsetminus \{0\}$. We denote by $\omega$ a Borel measure supported on $\mathcal{X}$, and we denote by $\mathcal{L}_2(\omega)$ the Hilbert space of square integrable real-valued functions on $\mathcal{X}$ with respect to $\omega$, equipped with the inner product $\langle \cdot, \cdot \rangle_\omega$, and the associated norm $\|.\|_\omega$. For $N \in \mathbb{N}^*$, we denote by $\omega^{\otimes N}$ the tensor product of $\omega$ defined on $\mathcal{X}^N$. Moreover, we denote by $\mathcal{F}$ the RKHS associated to the kernel $k : \mathcal{X} \times \mathcal{X} \to \mathbb{R}$ that we assume to be continuous and satisfying the condition $\int_\mathcal{X} k(x,x)\mathrm{d}\omega(x) < +\infty$. In particular, we assume the Mercer decomposition

$$k(x,y) = \sum_{m \in \mathbb{N}^*} \sigma_m \phi_m(x)\phi_m(y), \tag{2}$$

to hold, where the convergence is pointwise, and $\sigma_m$ and $\phi_m$ are the corresponding eigenvalues and eigenfunctions of the integral operator $\mathbf{\Sigma}$ defined for $f \in \mathcal{L}_2(\omega)$ by

$$\mathbf{\Sigma}f(\cdot) = \int_\mathcal{X} k(\cdot,y)f(y)\mathrm{d}\omega(y). \tag{3}$$

We assume that the sequence $\boldsymbol{\sigma} = (\sigma_m)_{m \in \mathbb{N}^*}$ is non-increasing and its elements are non-vanishing, and we assume that the corresponding eigenfunctions $\phi_m$ are continuous. Note that the $\phi_m$ are normalized: $\|\phi_m\|_\omega = 1$ for $m \in \mathbb{N}^*$. In particular, $(\phi_m)_{m \in \mathbb{N}^*}$ is an o.n.b. of $\mathcal{L}_2(\omega)$, and every element $f \in \mathcal{F}$ satisfies

$$\sum_{m \in \mathbb{N}^*} \frac{\langle f, \phi_m \rangle_\omega^2}{\sigma_m} < +\infty. \tag{4}$$

Moreover, for every $N \in \mathbb{N}^*$, we denote by $\mathcal{E}_N$ the eigen-subspace of $\mathcal{L}_2(\omega)$ spanned by $\phi_1, \ldots, \phi_N$. For any kernel $\kappa : \mathcal{X} \times \mathcal{X} \to \mathbb{R}$, and for $\boldsymbol{x} \in \mathcal{X}^N$, we define the kernel matrix $\boldsymbol{\kappa}(\boldsymbol{x}) := (\kappa(x_i, x_j))_{i,j \in [N]} \in \mathbb{R}^{N \times N}$. Finally, we denote in bold fonts the corresponding kernel matrices: $\boldsymbol{K}(\boldsymbol{x})$ for the kernel $k$, $\boldsymbol{K}_N(\boldsymbol{x})$ for the kernel $k_N$, $\boldsymbol{K}_N^\perp(\boldsymbol{x})$ for the kernel $k_N^\perp$, $\boldsymbol{\kappa}(\boldsymbol{x})$ for the kernel $\kappa$... Similarly, for any function $\mu : \mathcal{X} \to \mathbb{R}$ and for $\boldsymbol{x} \in \mathcal{X}^N$, we define the vector of evaluations $\mu(\boldsymbol{x}) := (\mu(x_i))_{i \in [N]} \in \mathbb{R}^N$.

## 2   Related work

In the section, we review some results that are relevant to the contribution.

### 2.1   Ermakov-Zolotukhin quadrature

The quadrature rule proposed by Ermakov and Zolotukhin in [14] deals with integrals that write

$$\int_\mathcal{X} f(x)\phi_m(x)\mathrm{d}\omega(x), \tag{5}$$

where $f \in \mathcal{L}_2(\omega)$, and $(\phi_m)_{m \in \mathbb{N}^*}$ is an orthonormal family with respect to the measure $\omega$. Its construction goes as follows. Let $N \in \mathbb{N}^*$ and let $\boldsymbol{x} \in \mathcal{X}^N$ such that the matrix

$$\boldsymbol{\Phi}_N(\boldsymbol{x}) := (\phi_n(x_i))_{(n,i) \in [N] \times [N]}$$

is non-singular. For $n \in [N]$, define

$$I^{\text{EZ},n}(f) = \sum_{i \in [N]} \hat{w}_i^{\text{EZ},n} f(x_i), \tag{6}$$

where $\hat{w}^{\text{EZ},n} := (\hat{w}_i^{\text{EZ},n})_{i \in [N]} \in \mathbb{R}^N$ is given by $\hat{w}^{\text{EZ},n} = \Phi_N(x)^{-1} e_n$, with $e_n$ is the vector of $\mathbb{R}^N$ with the $n$-th coordinate is 1 and the rest are 0 [1]. We can prove easily that this quadrature is exact

$$\forall f \in \text{Span}(\phi_n)_{n \in [N]}, \quad \sum_{i \in [N]} \hat{w}_i^{\text{EZ},n} f(x_i) = \int_{\mathcal{X}} f(x) \phi_n(x) \, \mathrm{d}\omega(x). \tag{7}$$

A quadrature rule that satisfies (7) is typically called an *interpolatory quadrature rule*.

Now, if $f \notin \text{Span}(\phi_n)_{n \in [N]}$, the authors studied the expected value and the variance of $I^{\text{EZ},n}(f)$ when $x = (x_1, \ldots, x_N)$ is taken to be a random variable in $\mathcal{X}^N$ that follows the distribution of density

$$p_{\text{DPP}}(x_1, \ldots, x_N) := \frac{1}{N!} \text{Det}^2 \Phi_N(x), \tag{8}$$

with respect to the product measure $\omega^{\otimes N}$ defined on $\mathcal{X}^N$. As it was observed in [16], the nodes of the quadrature follow the distribution of the determinantal point process of reference measure $\omega$ and marginal kernel $\kappa_N$ defined by $\kappa_N(x, y) = \sum_{n \in [N]} \phi_n(x) \phi_n(y)$. We refer to [20] for further details on determinantal point processes. Now, we recall the main result of [14].

**Theorem 1.** *Let $x$ be a random subset of $\mathcal{X}$ that follows the distribution of DPP of kernel $\kappa_N$ and reference measure $\omega$. Let $f \in \mathcal{L}_2(\omega)$, and $n \in [N]$. Then*

$$\mathbb{E}_{\text{DPP}} I^{\text{EZ},n}(f) = \int_{\mathcal{X}} f(x) \phi_n(x) \mathrm{d}\omega(x), \tag{9}$$

*and*

$$\mathbb{V}_{\text{DPP}} I^{\text{EZ},n}(f) = \sum_{m \geq N+1} \langle f, \phi_m \rangle_\omega^2. \tag{10}$$

Theorem 1 shows that the $I^{\text{EZ},n}(f)$ is an unbiased estimator of $\int_{\mathcal{X}} f(x) \phi_n(x) \mathrm{d}\omega(x)$, and its variance depends on the coefficients $\langle f, \phi_m \rangle_\omega$ for $m \geq N+1$. Consequently, the expected squared error of the quadrature is equal to the variance of $I^{\text{EZ},n}(f)$ and it is given by

$$\mathbb{E}_{\text{DPP}} \left| \int_{\mathcal{X}} f(x) \phi_n(x) \mathrm{d}\omega(x) - \sum_{i \in [N]} \hat{w}_i^{\text{EZ},n} f(x_i) \right|^2 = \sum_{m \geq N+1} \langle f, \phi_m \rangle_\omega^2. \tag{11}$$

The identity (11) gives a theoretical guarantee for an interpolatory quadrature rule when the nodes follow the distribution of the DPP defined by (8). Compared to existing work on the literature [11, 22, 23, 24], (11) is generic and applies to any orthonormal family.

Now, we may observe that the expected squared error in (11) depends strongly on the function $f$. This makes the comparison between EZQ and other quadratures, based on some test function $f$, tricky: the choice of $f$ may favor (or disfavor) EZQ. In order to circumvent this difficulty, we suggest to study a figure of merit that is independent of the choice of the function $f$. This is possible using kernels through the study of the worst-case integration error on the unit ball of an RKHS. The definition of this quantity will be recalled in the following section.

## 2.2 The worst integration error in kernel quadrature

The use of the kernel framework in the context of numerical integration can be tracked back to the work of Hickernell [18, 19], who introduced the use of kernels to the quasi Monte Carlo community. Their use was popularized in the machine learning community by [28, 9]. In this framework, the quality of a quadrature is assessed by the worst-case integration error on the unit ball of an RKHS $\mathcal{F}$ associated to some kernel $k : \mathcal{X} \times \mathcal{X} \to \mathbb{R}_+$. This quantity is defined as follows

$$\sup_{\substack{f \in \mathcal{F}, \\ \|f\|_{\mathcal{F}} \leq 1}} \left| \int_{\mathcal{X}} f(x) g(x) \mathrm{d}\omega(x) - \sum_{i \in [N]} w_i f(x_i) \right|. \tag{12}$$

---

[1]The dependency of the vector $\hat{w}^{\text{EZ},n}$ on $x$ was dropped for simplicity.

This quantity reflects how good is the quadrature uniformly on the unit ball of $\mathcal{F}$. Interestingly, this quantity has a closed formula

$$\left\| \mu_g - \sum_{i \in [N]} w_i k(x_i, .) \right\|_{\mathcal{F}}, \tag{13}$$

where $\mu_g = \Sigma g$ is the so-called *embedding* of $g$ in the RKHS $\mathcal{F}$. We shall use in Section 3.2 the equivalent expression (13) of the worst-case integration error, to derive a closed formula of

$$\mathbb{E}_{\text{DPP}} \sup_{\substack{f \in \mathcal{F}, \\ \|f\|_{\mathcal{F}} \leq 1}} \left| \int_{\mathcal{X}} f(x) g(x) \mathrm{d}\omega(x) - \sum_{i \in [N]} \hat{w}_i^{\text{EZ},n} f(x_i) \right|^2. \tag{14}$$

By now, we observe that the weights $\hat{w}_i^{\text{EZ},n}$ of EZQ are non-optimal in the sense that they do not minimize (13). By definition, the *optimal kernel quadrature* for a given configuration $\boldsymbol{x}$, such that the kernel matrix $\boldsymbol{K}(\boldsymbol{x})$ is non-singular, is the quadrature with nodes the $x_i$ and weights the $\hat{w}_i$ that minimize (13). We specify in Section 3.1 the subtle difference between the quadrature of Ermakov and Zolotukhin and the optimal kernel quadrature. Before that, we review the existing constructions of the optimal kernel quadrature in the following section.

## 2.3 The design of the optimal kernel quadrature

The optimal kernel quadrature may be calculated numerically under the assumption that the matrix $\boldsymbol{K}(\boldsymbol{x})$ is non-singular. Indeed, for a given configuration of nodes $\boldsymbol{x} \in \mathcal{X}^N$, the square of (13) is quadratic on $\boldsymbol{w}$ and have a unique solution given by $\hat{\boldsymbol{w}}^{\text{OKQ},g} = \boldsymbol{K}(\boldsymbol{x})^{-1} \mu_g(\boldsymbol{x})$ [2]. In particular, the optimal mixture $\sum_{i \in [N]} \hat{w}_i^{\text{OKQ},g} k(x_i, .)$ takes the same values as $\mu_g$ on the nodes $x_i$: the optimal mixture interpolates the function $\mu_g$ on the configuration of nodes $\boldsymbol{x}$. At this level, $\boldsymbol{x}$ is still a degree of freedom and need to be designed. This task was tackled by different approaches. One approach consists on using adhoc designs for which a theoretical analysis of the convergence rate is possible. This is the case of, inter alia, the uniform grid in the periodic Sobolev space [6, 26], higher-order digital nets sequences in tensor products of Sobolev spaces [8], or tensor product of scaled Hermite roots in the RKHS defined by the Gaussian kernel [22]. Another approach consists on using a sequential algorithm to build up the configuration $\boldsymbol{x}$ [12, 13, 21, 7]. In general, each step of these greedy algorithms requires to solve a non-convex problem and costly approximations must be employed. Alternatively, random designs, based on determinantal point processes and their mixtures [3, 4], were shown to have strong theoretical guarantees and competitive empirical performances. More precisely, it was shown that if $\boldsymbol{x}$ follows the distribution of DPP of reference measure $\omega$ and marginal kernel $\kappa_N$, and if $g \in \mathcal{L}_2(\omega)$ such that $\|g\|_\omega \leq 1$, then

$$\mathbb{E}_{\text{DPP}} \left\| \mu_g - \sum_{i \in [N]} \hat{w}_i^{\text{OKQ},g} k(x_i, .) \right\|_{\mathcal{F}}^2 \leq 2\sigma_{N+1} + 2\Big( \sum_{n \in [N]} |\langle g, \phi_n \rangle_\omega| \Big)^2 N r_N, \tag{15}$$

where $r_N = \sum_{m \geq N+1} \sigma_m$ [2] (Theorem 4.8). However, numerical simulations suggest that the l.h.s. of (15) converges to 0 at the faster rate $\mathcal{O}(\sigma_{N+1})$, which corresponds to the best achievable rate according to [4]. This optimal rate was proved to be achieved, under some mild conditions on the eigenvalues $\sigma_n$, using the distribution of continuous volume sampling (CVS) [4]. This distribution is a mixture of determinantal point processes and is closely tied to the projection DPP used in [3] and comes with the following guarantee

$$\forall g \in \mathcal{L}_2(\omega), \ \mathbb{E}_{\text{CVS}} \left\| \mu_g - \sum_{i \in [N]} \hat{w}_i^{\text{OKQ},g} k(x_i, .) \right\|_{\mathcal{F}}^2 = \sum_{m \in \mathbb{N}^*} \langle g, \phi_n \rangle_\omega^2 \epsilon_m(N), \tag{16}$$

where $\epsilon_m(N) = \mathcal{O}(\sigma_{N+1})$ for every $m \in \mathbb{N}^*$, so that the expected squared worst-integration error of OKQ under the continuous volume sampling distribution scales as $\mathcal{O}(\sigma_{N+1})$ for every $g \in \mathcal{L}_2(\omega)$.

---

[2]The dependency of the vector $\hat{\boldsymbol{w}}^{\text{OKQ},g}$ on $\boldsymbol{x}$ was dropped for simplicity.

# 3 Main results

This section gathers the main contributions of this article. In Section 3.1, we prove that both EZQ and OKQ belong to a larger class of quadrature rules called kernel-based interpolation quadrature (KBIQ). In Section 3.2, we prove a close formula of the expected squared worst-case integration error of EZQ. In Section 3.3, we use Theorem 3 to improve on the existing theoretical guarantees of OKQ with DPPs.

## 3.1 Kernel-based interpolation quadrature

In this section, we define a new class of quadrature rules that extends both Ermakov-Zolotukhin quadrature and the optimal kernel quadrature. We start by the following observation: the weights $\hat{w}_i^{\mathrm{EZ},n}(\boldsymbol{x})$ of EZQ, defined in (6), writes as

$$\hat{\boldsymbol{w}}^{\mathrm{EZ},n}(\boldsymbol{x}) = \boldsymbol{\Phi}_N(\boldsymbol{x})^{-1}\boldsymbol{e}_n. \tag{17}$$

By observing that $\phi_n(\boldsymbol{x}) = \boldsymbol{\Phi}_N(\boldsymbol{x})^{\mathsf{T}}\boldsymbol{e}_n$, and $\boldsymbol{\kappa}_N(\boldsymbol{x}) = \boldsymbol{\Phi}_N(\boldsymbol{x})^{\mathsf{T}}\boldsymbol{\Phi}_N(\boldsymbol{x})$, we prove that

$$\hat{\boldsymbol{w}}^{\mathrm{EZ},n}(\boldsymbol{x}) = \boldsymbol{\kappa}_N(\boldsymbol{x})^{-1}\phi_n(\boldsymbol{x}), \tag{18}$$

Equivalently, we have $\phi_n(\boldsymbol{x}) = \boldsymbol{\kappa}_N(\boldsymbol{x})\hat{\boldsymbol{w}}^{\mathrm{EZ},n}(\boldsymbol{x})$. In other words, $\sum_{i\in[N]}\hat{w}_i^{\mathrm{EZ},n}(\boldsymbol{x})\kappa_N(x_i,.)$ takes the same values as $\phi_n$ on the nodes $x_i$: $\hat{\boldsymbol{w}}^{\mathrm{EZ},n}(\boldsymbol{x})$ is the vector resulting of the interpolation of $\phi_n$ by the kernel $\kappa_N$. From this observation, we define kernel-based interpolation quadrature as an extension of EZQ as follows: let $\boldsymbol{\gamma} := (\gamma_m)_{m\in\mathbb{N}^*}$ be a sequence of positive real numbers, and let $M \in \mathbb{N}^* \cup \{+\infty\}$. Define the kernel $\kappa^{\boldsymbol{\gamma},M}$ on $\mathcal{X} \times \mathcal{X}$ by

$$\forall x,y \in \mathcal{X}, \ \ \kappa^{\boldsymbol{\gamma},M}(x,y) = \sum_{m=1}^{M} \gamma_m \phi_m(x)\phi_m(y). \tag{19}$$

Now, starting from a configuration $\boldsymbol{x} \in \mathcal{X}^N$ such that $\mathrm{Det}\,\boldsymbol{\kappa}_N(\boldsymbol{x}) > 0$, we have $\mathrm{Det}\,\boldsymbol{\kappa}^{\boldsymbol{\gamma},M}(\boldsymbol{x}) > 0$ [3], and for a given $g \in \mathcal{L}_2(\omega)$, we define the vector of weights $\hat{\boldsymbol{w}}^{\boldsymbol{\gamma},M,g}(\boldsymbol{x}) \in \mathbb{R}^N$ by

$$\hat{\boldsymbol{w}}^{\boldsymbol{\gamma},M,g}(\boldsymbol{x}) = \boldsymbol{\kappa}^{\boldsymbol{\gamma},M}(\boldsymbol{x})^{-1}\mu_g^{\boldsymbol{\gamma},M}(\boldsymbol{x}), \tag{20}$$

where

$$\mu_g^{\boldsymbol{\gamma},M}(x) = \sum_{m=1}^{M} \gamma_m \langle g, \phi_m \rangle_\omega \phi_m(x). \tag{21}$$

We check again that $\sum_{i\in[N]}\hat{w}_i^{\boldsymbol{\gamma},M,g}\kappa^{\boldsymbol{\gamma},M}(x_i,.)$ takes the same values as $\mu_g^{\boldsymbol{\gamma},M}$ on the nodes $x_i$: the mixture interpolates $\mu_g^{\boldsymbol{\gamma},M}$ on the nodes $x_i$. Now, for a given $g \in \mathcal{L}_2(\omega)$, the vector of weights $\hat{\boldsymbol{w}}^{\boldsymbol{\gamma},M,g}(\boldsymbol{x})$ have two degrees of freedom: the sequence $\boldsymbol{\gamma}$ and the rank of the kernel $M$. These degrees of freedom may be mixed in a variety of ways to cover a large class of quadrature rules. In particular, we show in Section 3.1.1 that, for any sequence $\boldsymbol{\gamma}$, KBIQ is equivalent to EZQ when $M = N$, and we show in Section 3.1.2 that KBIQ is equivalent to OKQ when $M = +\infty$ and $\boldsymbol{\gamma} = \boldsymbol{\sigma}$; as summarized in Table 1. We may also consider $M$ to be finite but strictly larger than $N$. Yet, the theoretical analysis of these intermediate quadrature rules is beyond the scope of this work.

| Quadrature | $M$ | $\boldsymbol{\gamma}$ | $\mu_g^{\boldsymbol{\gamma},M}$ | $\kappa^{\boldsymbol{\gamma},M}$ |
|---|---|---|---|---|
| EZQ | $N$ | Any | $\sum_{n\in[N]} \gamma_n \langle g, \phi_n \rangle_\omega \phi_n$ | $\kappa^{\boldsymbol{\gamma},N}$ |
| EZQ | $N$ | $\gamma_m = 1$ | $g_N := \sum_{n\in[N]} \langle g, \phi_n \rangle_\omega \phi_n$ | $\kappa_N$ |
| ... | ... | ... | ... | ... |
| OKQ | $+\infty$ | $\boldsymbol{\sigma}$ | $\mu_g$ | $k$ |

Table 1: An overview of some examples of KBIQ with the corresponding couples $(\kappa^{\boldsymbol{\gamma},M}, \mu_g^{\boldsymbol{\gamma},M})$.

---

[3]See Appendix D.1. in [3] for a proof.

### 3.1.1 EZQ is a special case of KBIQ

We recover EZQ, as defined in [14, 16], by taking $M = N$, and $\boldsymbol{\gamma}$ is defined by $\gamma_m = 1$ for every $m \in \mathbb{N}^*$, and $g \equiv \phi_n$ for some $n \in [N]$. The equivalent definition (20) extends EZQ to the situation when $g \notin \mathcal{E}_N$. Even better, we show in the following that $\hat{\boldsymbol{w}}^{\gamma,N,g}(\boldsymbol{x})$ is independent of $\boldsymbol{\gamma}$ when $M = N$. In particular, for any sequence of positive numbers $\boldsymbol{\gamma}$ we have

$$\forall n \in [N], \ \hat{\boldsymbol{w}}^{\gamma,N,\phi_n}(\boldsymbol{x}) = \hat{\boldsymbol{w}}^{\mathrm{EZ},n}(\boldsymbol{x}). \tag{22}$$

**Proposition 2.** *Let* $g \in \mathcal{E}_N$, *and let* $\boldsymbol{x} \in \mathcal{X}^N$ *such that* $\mathrm{Det}\, \boldsymbol{\kappa}_N(\boldsymbol{x}) > 0$. *Let* $\boldsymbol{\gamma} = (\gamma_m)_{m \in \mathbb{N}^*}$ *and* $\tilde{\boldsymbol{\gamma}} = (\tilde{\gamma}_m)_{m \in \mathbb{N}^*}$ *be two sequences of positive numbers. We have*

$$\hat{\boldsymbol{w}}^{\gamma,N,g}(\boldsymbol{x}) = \hat{\boldsymbol{w}}^{\tilde{\gamma},N,g}(\boldsymbol{x}). \tag{23}$$

Thanks to the invariance of $\hat{\boldsymbol{w}}^{\gamma,N,g}$ with respect to $\boldsymbol{\gamma}$, we simplify the notation and we write $\hat{\boldsymbol{w}}^{\mathrm{EZ},g}$ instead [4]. Moreover, using this invariance, EZQ may be seen as an approximation of OKQ when $g \in \mathcal{E}_N$. Indeed, by approximating the kernel matrix $\boldsymbol{K}(\boldsymbol{x}) \approx \boldsymbol{K}_N(\boldsymbol{x})$ where $k_N(x,y) = \sum_{n \in [N]} \sigma_n \phi_n(x)\phi_n(y)$, we have

$$\boldsymbol{K}(\boldsymbol{x})^{-1}\mu_g(\boldsymbol{x}) \approx \hat{\boldsymbol{w}}^{\mathrm{EZ},g}, \tag{24}$$

since $\boldsymbol{K}_N(\boldsymbol{x})^{-1}\mu_g(\boldsymbol{x}) = \hat{\boldsymbol{w}}^{\mathrm{EZ},g}$ by Proposition 2. Interestingly, this approximation is reminiscent to the one used in [23] in the case of the Gaussian kernel.

### 3.1.2 OKQ is a special case of KBIQ

The optimal kernel quadrature is a special case of KBIQ when $M = +\infty$ and $\boldsymbol{\gamma} = \boldsymbol{\sigma}$. Indeed, in this case, we have $\kappa^{\gamma,M} = k$, and $\mu_g^{\gamma,M} = \mu_g$, so that

$$\hat{\boldsymbol{w}}^{\sigma,M,g}(\boldsymbol{x}) = \boldsymbol{K}(\boldsymbol{x})^{-1}\mu_g(\boldsymbol{x}) = \hat{\boldsymbol{w}}^{\mathrm{OKQ},g}. \tag{25}$$

In other words, Ermakov-Zolotukhin quadrature and the optimal kernel quadrature are extreme instances of interpolation based kernel quadrature that correspond to the regimes $M = N$ and $M = +\infty$. As it was shown in Proposition 2, the weights of EZQ depend only on the eigenfunctions $\phi_m$ and do not depend on the eigenvalues $\sigma_m$. This is to be compared to the weights of OKQ that depend simultaneously on the eigenvalues and the eigenfunctions.

## 3.2 Main theorem

We give in this section the theoretical analysis of the worst case integration error of EZQ under the distribution of the projection DPP.

**Theorem 3.** *Let* $N \in \mathbb{N}^*$. *We have*

$$\forall g \in \mathcal{E}_N, \ \mathbb{E}_{\mathrm{DPP}} \|\mu_g - \sum_{i \in [N]} \hat{w}_i^{\mathrm{EZ},g} k(x_i, .)\|_{\mathcal{F}}^2 = \sum_{n \in [N]} \langle g, \phi_n \rangle_\omega^2 r_N, \tag{26}$$

*where* $r_N = \sum_{m \geq N+1} \sigma_m$. *Moreover,*

$$\forall g \in \mathcal{L}_2(\omega), \ \mathbb{E}_{\mathrm{DPP}} \|\mu_g - \sum_{i \in [N]} \hat{w}_i^{\mathrm{EZ},g} k(x_i, .)\|_{\mathcal{F}}^2 \leq 4\|g\|_\omega^2 r_N. \tag{27}$$

As an immediate consequence of Theorem 3, we have

$$\forall g \in \mathcal{L}_2(\omega), \ \mathbb{E}_{\mathrm{DPP}} \sup_{\substack{f \in \mathcal{F}, \\ \|f\|_{\mathcal{F}}=1}} \left| \int_{\mathcal{X}} f(x)g(x)\mathrm{d}\omega(x) - \sum_{i \in [N]} \hat{w}_i^{\mathrm{EZ},g} f(x_i) \right|^2 = \mathcal{O}(r_N). \tag{28}$$

In other words, the squared worst-case integration error of Ermakov-Zolotukhin quadrature with DPP nodes converges to 0 at the rate $\mathcal{O}(r_{N+1})$. This rate is slower than the rate of convergence of

---

[4] In the case $g \equiv \phi_n$ for some $n \in [N]$, we use $\hat{\boldsymbol{w}}^{\mathrm{EZ},\phi_n}$ or $\hat{\boldsymbol{w}}^{\mathrm{EZ},n}$ alternatively.

$\mathbb{E}_{\mathrm{DPP}} I^{\mathrm{EZ},n}(f)^2$ given by Theorem 1. Indeed, if $f \in \mathcal{F}$ then $\|f\|_{\mathcal{F}}^2 = \sum_{m \in \mathbb{N}^*} \langle f, \phi_m \rangle_\omega^2 / \sigma_m < +\infty$, and Theorem 1 yields

$$\mathbb{V}_{\mathrm{DPP}} I^{\mathrm{EZ},n}(f)^2 \leq \sigma_{N+1} \|f\|_{\mathcal{F}}^2, \tag{29}$$

so that

$$\mathbb{E}_{\mathrm{DPP}} \left| \int_{\mathcal{X}} f(x)\phi_n(x)\mathrm{d}\omega(x) - \sum_{i \in [N]} \hat{w}_i^{\mathrm{EZ},n} f(x_i) \right|^2 = \mathbb{V}_{\mathrm{DPP}} I^{\mathrm{EZ},n}(f)^2 = \mathcal{O}(\sigma_{N+1}). \tag{30}$$

Now, observe that for some sequences we have $\sigma_{N+1} = o(r_{N+1})$. For instance, if $\sigma_m = m^{-2s}$ for some $s > 1/2$, then $r_{N+1} = \mathcal{O}(N^{1-2s})$. We conclude that the convergence of EZQ under DPP is slower than the optimal rate $\mathcal{O}(\sigma_{N+1})$, that was observed empirically for OKQ under DPP in [3] and proved theoretically for OKQ under CVS in [4], if we consider the worst-case integration error as a figure of merit. This is to be compared with the theoretical result of [14] that can not predict the difference in the rate of convergence between EZQ and OKQ: our analysis highlights the interest of using kernels when comparing quadratures.

## 3.3 Improved theoretical guarantees for the optimal kernel quadrature with DPPs

Theorem 3 improves on the existing theoretical guarantees of the optimal kernel quadrature with determinantal point processes initially proposed in [3]. This is the purpose of the following result.

**Theorem 4.** *Let $N \in \mathbb{N}^*$. We have*

$$\forall g \in \mathcal{L}_2(\omega), \ \mathbb{E}_{\mathrm{DPP}} \|\mu_g - \sum_{i \in [N]} \hat{w}_i^{\mathrm{OKQ},g} k(x_i, .)\|_{\mathcal{F}}^2 \leq 4\|g\|_\omega^2 r_N. \tag{31}$$

Compared to the analysis conducted in [3], Theorem 4 offers a sharper upper bound of

$$\mathbb{E}_{\mathrm{DPP}} \|\mu_g - \sum_{i \in [N]} \hat{w}_i^{\mathrm{OKQ},g} k(x_i, .)\|_{\mathcal{F}}^2. \tag{32}$$

Indeed, the upper bound (31) is dominated by $\sum_{n=1}^N \langle g, \phi_n \rangle_\omega^2 r_N$ comparably to the upper bound (15), proved in [3], dominated by $(\sum_{n=1}^N |\langle g, \phi_n \rangle_\omega|)^2 N r_N$: our bound improves upon (15) by a factor of $N^2$, since

$$(\sum_{n=1}^N |\langle g, \phi_n \rangle_\omega|)^2 \leq N \sum_{n=1}^N \langle g, \phi_n \rangle_\omega^2 \leq N\|g\|_\omega^2. \tag{33}$$

Theorem 4 follows immediately from Theorem 3 by observing that

$$\|\mu_g - \sum_{i \in [N]} \hat{w}_i^{\mathrm{OKQ},g} k(x_i, .)\|_{\mathcal{F}}^2 \leq \|\mu_g - \sum_{i \in [N]} \hat{w}_i^{\mathrm{EZ},g} k(x_i, .)\|_{\mathcal{F}}^2. \tag{34}$$

Table 2 summarizes the theoretical contributions of this work compared to the existing literature.

| Quadrature | Distribution | Theoretical rate | Empirical rate | Reference |
|---|---|---|---|---|
| EZQ | DPP | $\mathcal{O}(r_{N+1})$ | $\mathcal{O}(r_{N+1})$ | Theorem 3 |
| OKQ | DPP | $N^2 \mathcal{O}(r_{N+1})$ | $\mathcal{O}(\sigma_{N+1})$ | [3] |
| | | $\mathcal{O}(r_{N+1})$ | $\mathcal{O}(\sigma_{N+1})$ | Theorem 4 |
| OKQ | CVS | $\mathcal{O}(\sigma_{N+1})$ | $\mathcal{O}(\sigma_{N+1})$ | [4] |

Table 2: A comparison of the rates given by Theorem 3 and Theorem 4 compared to the existing guarantees in the literature.

We give in the following section, a sketch of the main ideas behind the proof of Theorem 3.

## 4 Sketch of the proof

The proof of Theorem 3 decomposes into two steps. First, in Section 4.1, we give a decomposition of the squared approximation error $\|\mu_g - \sum_{i \in [N]} \hat{w}_i^{\mathrm{EZ},g} k(x_i, .)\|_{\mathcal{F}}^2$, then, in Section 4.2, we use this decomposition to prove a closed formula of $\mathbb{E}_{\mathrm{DPP}} \|\mu_g - \sum_{i \in [N]} \hat{w}_i^{\mathrm{EZ},g} k(x_i, .)\|_{\mathcal{F}}^2$.

## 4.1 A decomposition of the approximation error

Let $g \in \mathcal{E}_N$ and let $\boldsymbol{x} \in \mathcal{X}^N$ such that $\mathrm{Det}\,\boldsymbol{\kappa}_N(\boldsymbol{x}) > 0$, we have

$$\|\mu_g - \sum_{i \in [N]} \hat{w}_i^{\mathrm{EZ},g} k(x_i, .)\|_{\mathcal{F}}^2 = \|\mu_g\|_{\mathcal{F}}^2 - 2\mu_g(\boldsymbol{x})^{\mathsf{T}}\hat{\boldsymbol{w}}^{\mathrm{EZ},g} + \hat{\boldsymbol{w}}^{\mathrm{EZ},g\mathsf{T}} \boldsymbol{K}(\boldsymbol{x})\hat{\boldsymbol{w}}^{\mathrm{EZ},g}. \tag{35}$$

The last two terms of the r.h.s of (35) decompose as follows.

**Proposition 5.** *Let $g \in \mathcal{E}_N$ and let $\boldsymbol{x} \in \mathcal{X}^N$ such that $\mathrm{Det}\,\boldsymbol{\kappa}_N(\boldsymbol{x}) > 0$. We have*

$$\mu_g(\boldsymbol{x})^{\mathsf{T}}\hat{\boldsymbol{w}}^{\mathrm{EZ},g} = \|\mu_g\|_{\mathcal{F}}^2, \tag{36}$$

*and*

$$\hat{\boldsymbol{w}}^{\mathrm{EZ},g\mathsf{T}} \boldsymbol{K}(\boldsymbol{x})\hat{\boldsymbol{w}}^{\mathrm{EZ},g} = \|\mu_g\|_{\mathcal{F}}^2 + \boldsymbol{\epsilon}^{\mathsf{T}}\boldsymbol{\Phi}_N(\boldsymbol{x})^{-1\mathsf{T}} \boldsymbol{K}_N^{\perp}(\boldsymbol{x})\boldsymbol{\Phi}_N(\boldsymbol{x})^{-1}\boldsymbol{\epsilon}, \tag{37}$$

*where $\boldsymbol{\epsilon} = \sum_{n \in [N]}\langle g, \phi_n\rangle_\omega \boldsymbol{e}_n$, and $k_N^{\perp}$ is the kernel defined by*

$$k_N^{\perp}(x, y) = \sum_{m \geq N+1} \sigma_m \phi_m(x)\phi_m(y). \tag{38}$$

The proof of Proposition 5 is detailed in Appendix A.3. Now, by combining (35), (36) and (37), we get

$$\|\mu_g - \sum_{i \in [N]} \hat{w}_i^{\mathrm{EZ},g} k(x_i, .)\|_{\mathcal{F}}^2 = \|\mu_g\|_{\mathcal{F}}^2 - 2\|\mu_g\|_{\mathcal{F}}^2 + \|\mu_g\|_{\mathcal{F}}^2 + \boldsymbol{\epsilon}^{\mathsf{T}}\boldsymbol{\Phi}_N(\boldsymbol{x})^{-1\mathsf{T}} \boldsymbol{K}_N^{\perp}(\boldsymbol{x})\boldsymbol{\Phi}_N(\boldsymbol{x})^{-1}\boldsymbol{\epsilon}.$$

This proves the following result.

**Theorem 6.** *Let $g \in \mathcal{E}_N$ and let $\boldsymbol{x} \in \mathcal{X}^N$ such that $\mathrm{Det}\,\boldsymbol{\kappa}_N(\boldsymbol{x}) > 0$. We have*

$$\|\mu_g - \sum_{i \in [N]} \hat{w}_i^{\mathrm{EZ},g} k(x_i, .)\|_{\mathcal{F}}^2 = \boldsymbol{\epsilon}^{\mathsf{T}}\boldsymbol{\Phi}_N(\boldsymbol{x})^{-1\mathsf{T}} \boldsymbol{K}_N^{\perp}(\boldsymbol{x})\boldsymbol{\Phi}_N(\boldsymbol{x})^{-1}\boldsymbol{\epsilon}, \tag{39}$$

*where $\boldsymbol{\epsilon} = \sum_{n \in [N]}\langle g, \phi_n\rangle_\omega \boldsymbol{e}_n$.*

## 4.2 A tractable formula of the expected approximation error

In the following, we prove a closed formula for $\mathbb{E}_{\mathrm{DPP}}\|\mu_g - \sum_{i \in [N]} \hat{w}_i^{\mathrm{EZ},g} k(x_i, .)\|_{\mathcal{F}}^2$. By Theorem 6, it is enough to calculate

$$\mathbb{E}_{\mathrm{DPP}}\boldsymbol{\epsilon}^{\mathsf{T}}\boldsymbol{\Phi}_N(\boldsymbol{x})^{-1\mathsf{T}} \boldsymbol{K}_N^{\perp}(\boldsymbol{x})\boldsymbol{\Phi}_N(\boldsymbol{x})^{-1}\boldsymbol{\epsilon}, \tag{40}$$

for $\boldsymbol{\epsilon} \in \mathbb{R}^N$. For this purpose, observe that $\boldsymbol{K}_N^{\perp}(\boldsymbol{x}) = \sum_{m \geq N+1} \sigma_m \phi_m(\boldsymbol{x})\phi_m(\boldsymbol{x})^{\mathsf{T}}$, so that

$$\boldsymbol{\epsilon}^{\mathsf{T}}\boldsymbol{\Phi}_N(\boldsymbol{x})^{-1\mathsf{T}} \boldsymbol{K}_N^{\perp}(\boldsymbol{x})\boldsymbol{\Phi}_N(\boldsymbol{x})^{-1}\boldsymbol{\epsilon} = \sum_{m \geq N+1} \sigma_m \boldsymbol{\epsilon}^{\mathsf{T}}\boldsymbol{\Phi}_N(\boldsymbol{x})^{-1\mathsf{T}} \phi_m(\boldsymbol{x})\phi_m(\boldsymbol{x})^{\mathsf{T}}\boldsymbol{\Phi}_N(\boldsymbol{x})^{-1}\boldsymbol{\epsilon}. \tag{41}$$

Therefore, the calculation of 40 boils down to the calculation of

$$\mathbb{E}_{\mathrm{DPP}}\boldsymbol{\epsilon}^{\mathsf{T}}\boldsymbol{\Phi}_N(\boldsymbol{x})^{-1\mathsf{T}} \phi_m(\boldsymbol{x})\phi_m(\boldsymbol{x})^{\mathsf{T}}\boldsymbol{\Phi}_N(\boldsymbol{x})^{-1}\boldsymbol{\epsilon}, \tag{42}$$

for $m \geq N+1$. This is the purpose of the following result.

**Theorem 7.** *Let $\boldsymbol{\epsilon} = \sum_{n \in [N]} \epsilon_n \boldsymbol{e}_n, \tilde{\boldsymbol{\epsilon}} = \sum_{n \in [N]} \tilde{\epsilon}_n \boldsymbol{e}_n \in \mathbb{R}^N$, and $m \geq N+1$. Then*

$$\mathbb{E}_{\mathrm{DPP}}\boldsymbol{\epsilon}^{\mathsf{T}}\boldsymbol{\Phi}_N(\boldsymbol{x})^{-1\mathsf{T}} \phi_m(\boldsymbol{x})\phi_m(\boldsymbol{x})^{\mathsf{T}}\boldsymbol{\Phi}_N(\boldsymbol{x})^{-1}\tilde{\boldsymbol{\epsilon}} = \sum_{n \in [N]} \epsilon_n \tilde{\epsilon}_n. \tag{43}$$

*In particular,*

$$\mathbb{E}_{\mathrm{DPP}}\boldsymbol{\epsilon}^{\mathsf{T}}\boldsymbol{\Phi}_N(\boldsymbol{x})^{-1\mathsf{T}} \boldsymbol{K}_N^{\perp}(\boldsymbol{x})\boldsymbol{\Phi}_N(\boldsymbol{x})^{-1}\tilde{\boldsymbol{\epsilon}} = \sum_{m \geq N+1} \sigma_m \sum_{n \in [N]} \epsilon_n \tilde{\epsilon}_n. \tag{44}$$

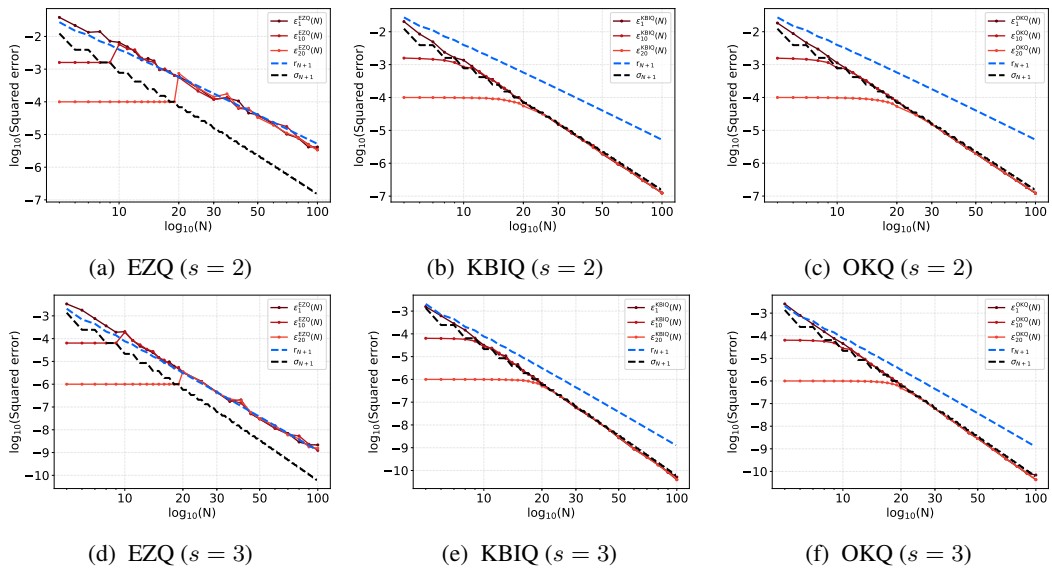

Figure 1: Squared worst-case integration error vs. number of nodes $N$ for EZQ, KBIQ and OKQ in the Sobolev space of periodic functions of order $s \in \{2, 3\}$.

We give the proof of Theorem 7 in Appendix A.4. By taking $\epsilon = \tilde{\epsilon} = \sum_{n \in [N]} \langle g, \phi_n \rangle_\omega e_n$ in Theorem 7, we obtain (26). As for (27), it is sufficient to observe that

$$\|\mu_g - \sum_{i \in [N]} \hat{w}_i^{\text{EZ},g} k(x_i, .)\|_{\mathcal{F}}^2 \leq 2 \Big( \|\mu_g - \mu_{g_N}\|_{\mathcal{F}}^2 + \|\mu_{g_N} - \sum_{i \in [N]} \hat{w}_i^{\text{EZ},g} k(x_i, .)\|_{\mathcal{F}}^2 \Big), \qquad (45)$$

where $g_N = \sum_{n \in [N]} \langle g, \phi_n \rangle_\omega \phi_n \in \mathcal{E}_N$, so that we can apply (26) to $g_N$ and we obtain

$$\mathbb{E}_{\text{DPP}} \|\mu_{g_N} - \sum_{i \in [N]} \hat{w}_i^{\text{EZ},g} k(x_i, .)\|_{\mathcal{F}}^2 = \sum_{n \in [N]} \langle g, \phi_n \rangle_\omega^2 \sum_{m \geq N+1} \sigma_m \leq \|g\|_\omega^2 \sum_{m \geq N+1} \sigma_m. \qquad (46)$$

The term $\|\mu_g - \mu_{g_N}\|_{\mathcal{F}}^2$ is upper bounded by $\sigma_{N+1} \|g\|_\omega^2$. We give the details in Appendix A. This concludes the proof of Theorem 3. In the following section, we give numerical experiments illustrating this result.

## 5 Numerical experiments

In this section, we illustrate the theoretical results presented in Section 3 in the case of the RKHS associated to the kernel

$$k_s(x, y) = 1 + \sum_{m \in \mathbb{N}^*} \frac{1}{m^{2s}} \cos(2\pi m(x - y)), \qquad (47)$$

that corresponds to the periodic Sobolev space of order $s$ on $[0, 1]$ [5], and we take $\omega$ to be the uniform measure on $\mathcal{X} = [0, 1]$. We compare the squared worst-case integration error of EZQ and OKQ and KBIQ, with $M = 2N$ and $\gamma = \sigma$, for $x$ that follows the distribution of the projection DPP and for $g \in \{e_1, e_{10}, e_{20}\}$. We take $N \in [5, 100]$. Figure 1 shows log-log plots of the squared error w.r.t. $N$, averaged over 1000 samples for each point, for $s \in \{2, 3\}$. We observe that the squared error of EZQ converges to 0 at the exact rate $\mathcal{O}(r_{N+1})$ predicted by Theorem 3, while the squared error of OKQ converges to 0 at the rate $\mathcal{O}(\sigma_{N+1})$ as it was already observed in [3], which is still better than the rate $\mathcal{O}(r_{N+1})$ proved in Theorem 4. Finally, KBIQ ($M = 2N$ and $\gamma = \sigma$) converges to 0 at the rate $\mathcal{O}(\sigma_{N+1})$. We conclude that, by taking $M = \alpha N$ with $\alpha > 1$, KBIQ have practically the same averaged error as OKQ ($M = +\infty$). As we have mentioned before, the theoretical analysis of KBIQ in the regime when $M$ is finite and strictly larger than $N$ is beyond the scope of this work, and we defer it for future work.

# 6 Conclusion

We studied the quadrature rule proposed by Ermakov and Zolotukhin through the lens of kernel methods. We proved that EZQ and OKQ belong to a larger class of quadrature rules that may be defined through kernel based interpolation. From this new perspective, EZQ may be seen as an approximation of OKQ. Moreover, we studied the expected value of the squared worst-case integration error of EZQ when the nodes follow the distribution of a DPP. In particular, we proved that EZQ converges to $0$ at the rate $\mathcal{O}(r_{N+1})$ which is slower than the optimal rate $\mathcal{O}(\sigma_{N+1})$ typically observed for OKQ with DPPs. This work shows the importance of the worst-case integration error as a figure of merit when comparing quadrature rules. Interestingly, we use our analysis of EZQ to improve upon the existing theoretical guarantees of OKQ under DPPs. Finally, we illustrated the theoretical results by some numerical experiments that hint that KBIQ in the regime $M > N$ may have similar performances as OKQ. It would be interesting to study this broader class of quadratures in the future.

## Broader impact

This article makes contributions to the fundamentals of numerical integration, and due to its theoretical nature, the author sees no ethical or immediate societal consequence of this work.

## Acknowledgments and Disclosure of Funding

This project was supported by the AllegroAssai ANR project ANR-19-CHIA-0009. The author would like to thank the reviewers for their thorough and constructive reviews and would like to thank Pierre Chainais and Rémi Bardenet for their constructive feedback on an early version of this work.

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
