# Supplementary material for
# *An analysis of Ermakov-Zolotukhin quadrature using kernels*

**Ayoub Belhadji**
Univ Lyon, ENS de Lyon
Inria, CNRS, UCBL
LIP UMR 5668, Lyon, France
ayoub.belhadji@ens-lyon.fr

## A   Detailed proofs

### A.1   Proof of Proposition 2

By definition, we have

$$\forall x_1, x_2 \in \mathcal{X}, \ \ \kappa_N^{\gamma}(x_1, x_2) = \sum_{n \in [N]} \gamma_n \phi_n(x_1) \phi_n(x_2), \tag{1}$$

therefore

$$\forall x_1, x_2 \in \mathcal{X}, \ \ \kappa_N^{\gamma}(x_1, x_2) = \sum_{n \in [N]} \rho_n \phi_n(x_1) \tilde{\gamma}_n \phi_n(x_2), \tag{2}$$

where

$$\forall n \in [N], \ \ \rho_n = \frac{\gamma_n}{\tilde{\gamma}_n}. \tag{3}$$

Then

$$\forall \boldsymbol{x} \in \mathcal{X}^N, \ \ \boldsymbol{\kappa}_N^{\gamma}(\boldsymbol{x}) = \boldsymbol{\Phi}_N^{\rho}(\boldsymbol{x})^{\intercal} \boldsymbol{\Phi}_N^{\tilde{\gamma}}(\boldsymbol{x}) \ , \tag{4}$$

where

$$\boldsymbol{\Phi}_N^{\rho}(\boldsymbol{x}) = (\rho_n \phi_n(x_i))_{(n,i) \in [N] \times [N]} \in \mathbb{R}^{N \times N}, \tag{5}$$

and

$$\boldsymbol{\Phi}_N^{\tilde{\gamma}}(\boldsymbol{x}) = (\tilde{\gamma}_n \phi_n(x_i))_{(n,i) \in [N] \times [N]} \in \mathbb{R}^{N \times N}. \tag{6}$$

Moreover, by definition of $\mu_g^{\gamma}$, we have

$$\forall x \in \mathcal{X}, \ \ \mu_g^{\gamma}(x) = \sum_{n \in [N]} \gamma_n \langle g, \phi_n \rangle_{\omega} \phi_n(x), \tag{7}$$

therefore

$$\forall x \in \mathcal{X}, \ \ \mu_g^{\gamma}(x) = \sum_{n \in [N]} \tilde{\gamma}_n \langle g, \phi_n \rangle_{\omega} \rho_n \phi_n(x), \tag{8}$$

so that

$$\forall \boldsymbol{x} \in \mathcal{X}^N, \ \ \mu_g^{\gamma}(\boldsymbol{x}) = \boldsymbol{\Phi}_N^{\rho}(\boldsymbol{x})^{\intercal} \boldsymbol{\alpha}, \tag{9}$$

where

$$\boldsymbol{\alpha} = (\tilde{\gamma}_n)_{n \in [N]} \in \mathbb{R}^N. \tag{10}$$

35th Conference on Neural Information Processing Systems (NeurIPS 2021).

Combining (4) and (9), we prove that for any $x \in \mathcal{X}^N$ such that $\mathrm{Det}\, \kappa_N(x) > 0$, we have

$$
\begin{aligned}
\hat{w}^{\gamma, N, g}(x) &= \kappa_N^\gamma(x)^{-1} \mu_g^\gamma(x) \\
&= \Phi_N^{\tilde{\gamma}}(x)^{-1}\ \Phi_N^\rho(x)^{-1\mathsf{T}} \mu_g^\gamma(x) \\
&= \Phi_N^{\tilde{\gamma}}(x)^{-1}\ \Phi_N^\rho(x)^{-1\mathsf{T}} \Phi_N^\rho(x)^\mathsf{T} \alpha \\
&= \Phi_N^{\tilde{\gamma}}(x)^{-1}\ \alpha.
\end{aligned}
\tag{11}
$$

## A.2 Useful results

We gather in this section some results that will be useful in the following proofs.

### A.2.1 A useful lemma

We prove in the following a lemma that we will use in Section A.3.

**Lemma 1.** *Let* $x \in \mathcal{X}^N$ *such that* $\mathrm{Det}\, \kappa_N(x) > 0$. *For* $n, n' \in [N]$, *define*

$$
\tau_{n,n'}(x) = \sqrt{\sigma_n}\sqrt{\sigma_{n'}} \phi_n(x)^\mathsf{T} K_N(x)^{-1} \phi_{n'}(x).
\tag{12}
$$

*Then*

$$
\forall n, n' \in [N], \ \tau_{n,n'}(x) = \delta_{n,n'}.
\tag{13}
$$

*Proof.* We have

$$
K_N(x) = \Phi_N^{\sqrt{\sigma}}(x)^\mathsf{T} \Phi_N^{\sqrt{\sigma}}(x),
\tag{14}
$$

where

$$
\Phi_N^{\sqrt{\sigma}}(x) = (\sqrt{\sigma}_n \phi_n(x_i))_{(n,i) \in [N] \times [N]}.
\tag{15}
$$

Let $n, n' \in [N]$. We have

$$
\sqrt{\sigma_n} \phi_n(x)^\mathsf{T} = e_n^\mathsf{T} \Phi_N^\sigma(x),
\tag{16}
$$

and

$$
\sqrt{\sigma_{n'}} \phi_{n'}(x) = \Phi_N^\sigma(x)^\mathsf{T} e_{n'}.
\tag{17}
$$

Therefore

$$
\begin{aligned}
\tau_{n,n'}(x) &= e_n^\mathsf{T} \Phi_N^\sigma(x) \left( \Phi_N^\sigma(x)^\mathsf{T} \Phi_N^\sigma(x) \right)^{-1} \Phi_N^\sigma(x)^\mathsf{T} e_{n'} \\
&= e_n^\mathsf{T} \Phi_N^\sigma(x) \Phi_N^\sigma(x)^{-1} \Phi_N^\sigma(x)^{\mathsf{T}^{-1}} \Phi_N^\sigma(x)^\mathsf{T} e_{n'} \\
&= e_n^\mathsf{T} e_{n'} \\
&= \delta_{n,n'}.
\end{aligned}
\tag{18}\tag{19}\tag{20}\tag{21}
$$

$\square$

### A.2.2 A borrowed result

We recall in the following a result proven in [1] that we will use in Section A.4.

**Proposition 1.** *[Theorem 1 in [1]] Let* $x$ *be a random subset of* $\mathcal{X}$ *that follows the distribution of DPP of kernel* $\kappa_N$ *and reference measure* $\omega$. *Let* $f \in \mathcal{L}_2(\omega)$, *and* $n, n' \in [N]$ *such that* $n \neq n'$. *Then*

$$
\mathbb{C}\mathrm{ov}_{\mathrm{DPP}}(I^{\mathrm{EZ},n}(f), I^{\mathrm{EZ},n'}(f)) = 0.
\tag{22}
$$

## A.3 Proof of Proposition 5

Let $x \in \mathcal{X}^N$ such that the condition $\mathrm{Det}\, \kappa_N(x) > 0$ is satisfied, and let $g \in \mathcal{E}_N$. We start by the proof of (36). By definition of $\mu_g$, we have

$$
\mu_g(x) = \sum_{n \in [N]} \sigma_n \langle g, \phi_n \rangle_\omega \phi_n(x),
\tag{23}
$$

so that

$$
\mu_g(x) = \sum_{n \in [N]} \sigma_n \langle g, \phi_n \rangle_\omega \phi_n(x).
\tag{24}
$$

Proposition 2 yields

$$\hat{\boldsymbol{w}}^{\mathrm{EZ},g} = \boldsymbol{K}_N(\boldsymbol{x})^{-1}\mu_g(\boldsymbol{x}). \tag{25}$$

Therefore

$$
\begin{aligned}
\hat{\boldsymbol{w}}^{\mathrm{EZ},g\mathsf{T}}\mu_g(\boldsymbol{x}) &= \mu_g(\boldsymbol{x})\boldsymbol{K}_N(\boldsymbol{x})^{-1}\mu_g(\boldsymbol{x}) \\
&= \sum_{n,n'\in[N]} \sigma_n\sigma_{n'}\langle g,\phi_n\rangle_\omega\langle g,\phi_{n'}\rangle_\omega\phi_n(\boldsymbol{x})^\mathsf{T}\boldsymbol{K}_N(\boldsymbol{x})^{-1}\phi_{n'}(\boldsymbol{x}) \\
&= \sum_{n\in[N]} \sigma_n\langle g,\phi_n\rangle_\omega^2\tau_{n,n}(\boldsymbol{x}) \\
&\quad + \sum_{\substack{n,n'\in[N]\\ n\neq n'}} \sqrt{\sigma_n\sigma_{n'}}\langle g,\phi_n\rangle_\omega\langle g,\phi_{n'}\rangle_\omega\tau_{n,n'}(\boldsymbol{x}),
\end{aligned}
\tag{26}
$$

where $\tau_{n,n'}(\boldsymbol{x})$ is defined by

$$\tau_{n,n'}(\boldsymbol{x}) = \sqrt{\sigma_n}\sqrt{\sigma_{n'}}\phi_n(\boldsymbol{x})^\mathsf{T}\boldsymbol{K}_N(\boldsymbol{x})^{-1}\phi_{n'}(\boldsymbol{x}). \tag{27}$$

Now, Lemma 1 yields

$$
\begin{aligned}
\sum_{n\in[N]} \sigma_n\langle g,\phi_n\rangle_\omega^2\tau_{n,n}(\boldsymbol{x}) &= \sum_{n\in[N]} \sigma_n\langle g,\phi_n\rangle_\omega^2 \\
&= \|\mu_g\|_{\mathcal{F}}^2,
\end{aligned}
\tag{28}
$$

and

$$\sum_{\substack{n,n'\in[N]\\ n\neq n'}} \sqrt{\sigma_n}\sqrt{\sigma_{n'}}\langle g,\phi_n\rangle_\omega\langle g,\phi_{n'}\rangle_\omega\tau_{n,n'}(\boldsymbol{x}) = 0. \tag{29}$$

Combining (26), (28) and (29), we obtain

$$\hat{\boldsymbol{w}}^{\mathrm{EZ},g\mathsf{T}}\mu_g(\boldsymbol{x}) = \|\mu_g\|_{\mathcal{F}}^2. \tag{30}$$

We move now to the proof of (37). We have by the Mercer decomposition

$$\boldsymbol{K}(\boldsymbol{x}) = \sum_{m=1}^{+\infty} \sigma_m\phi_m(\boldsymbol{x})\phi_m(\boldsymbol{x})^\mathsf{T} \tag{31}$$

$$= \sum_{m=1}^{N} \sigma_m\phi_m(\boldsymbol{x})\phi_m(\boldsymbol{x})^\mathsf{T} + \sum_{m=N+1}^{+\infty} \sigma_m\phi_m(\boldsymbol{x})\phi_m(\boldsymbol{x})^\mathsf{T}. \tag{32}$$

Moreover, observe that

$$\boldsymbol{K}_N(\boldsymbol{x}) = \sum_{m=1}^{N} \sigma_m\phi_m(\boldsymbol{x})\phi_m(\boldsymbol{x})^\mathsf{T}, \tag{33}$$

and

$$\boldsymbol{K}_N^\perp(\boldsymbol{x}) = \sum_{m=N+1}^{+\infty} \sigma_m\phi_m(\boldsymbol{x})\phi_m(\boldsymbol{x})^\mathsf{T}. \tag{34}$$

Therefore

$$\boldsymbol{K}(\boldsymbol{x}) = \boldsymbol{K}_N(\boldsymbol{x}) + \boldsymbol{K}_N^\perp(\boldsymbol{x}), \tag{35}$$

so that

$$\hat{\boldsymbol{w}}^{\mathrm{EZ},g\mathsf{T}}\boldsymbol{K}(\boldsymbol{x})\hat{\boldsymbol{w}}^{\mathrm{EZ},g} = \hat{\boldsymbol{w}}^{\mathrm{EZ},g\mathsf{T}}\boldsymbol{K}_N(\boldsymbol{x})\hat{\boldsymbol{w}}^{\mathrm{EZ},g} + \hat{\boldsymbol{w}}^{\mathrm{EZ},g\mathsf{T}}\boldsymbol{K}_N^\perp(\boldsymbol{x})\hat{\boldsymbol{w}}^{\mathrm{EZ},g}. \tag{36}$$

In order to evaluate $\hat{\boldsymbol{w}}^{\mathrm{EZ},g\mathsf{T}}\boldsymbol{K}_N(\boldsymbol{x})\hat{\boldsymbol{w}}^{\mathrm{EZ},g}$, we use Proposition 2, and we get

$$\hat{\boldsymbol{w}}^{\mathrm{EZ},g} = \boldsymbol{K}_N(\boldsymbol{x})^{-1}\mu_g(\boldsymbol{x}), \tag{37}$$

so that

$$
\begin{aligned}
\hat{\boldsymbol{w}}^{\mathrm{EZ},g\mathsf{T}}\boldsymbol{K}_N(\boldsymbol{x})\hat{\boldsymbol{w}}^{\mathrm{EZ},g} &= \hat{\boldsymbol{w}}^{\mathrm{EZ},g\mathsf{T}}\boldsymbol{K}_N(\boldsymbol{x})\boldsymbol{K}_N(\boldsymbol{x})^{-1}\mu_g(\boldsymbol{x}) \\
&= \hat{\boldsymbol{w}}^{\mathrm{EZ},g\mathsf{T}}\mu_g(\boldsymbol{x}) \\
&= \|\mu_g\|_{\mathcal{F}}^2.
\end{aligned}
\tag{38}
$$

Finally, by definition

$$\hat{w}^{\text{EZ},g} = \Phi_N(x)^{-1}\epsilon, \tag{39}$$

where $\epsilon = \sum_{n\in[N]}\langle g,\phi_n\rangle_\omega e_n$. Therefore

$$\hat{w}^{\text{EZ},g\mathsf{T}}K_N(x)^\perp \hat{w}^{\text{EZ},g} = \epsilon^\mathsf{T}\Phi_N(x)^{-1\mathsf{T}}K_N(x)^\perp\Phi_N(x)^{-1}\epsilon. \tag{40}$$

## A.4 Proof of Theorem 7

Let $m \in \mathbb{N}^*$ such that $m \geq N+1$. We prove that

$$\forall\epsilon,\tilde{\epsilon}\in\mathbb{R}^N,\ \ \mathbb{E}_{\text{DPP}}\epsilon^\mathsf{T}\Phi_N(x)^{-1\mathsf{T}}\phi_m(x)\phi_m(x)^\mathsf{T}\Phi_N(x)^{-1}\tilde{\epsilon} = \sum_{n\in N}\epsilon_n\tilde{\epsilon}_n. \tag{41}$$

For this purpose, let $\epsilon,\tilde{\epsilon}\in\mathbb{R}^N$, and observe that

$$\epsilon^\mathsf{T}\Phi_N(x)^{-1\mathsf{T}}\phi_m(x) = \sum_{n\in[N]}\epsilon_n e_n^\mathsf{T}\Phi_N(x)^{-1\mathsf{T}}\phi_m(x) \tag{42}$$

$$= \sum_{n\in[N]}\epsilon_n\hat{w}^{\text{EZ},n}\phi_m(x) \tag{43}$$

$$= \sum_{n\in[N]}\epsilon_n I^{\text{EZ},n}(\phi_m). \tag{44}$$

and

$$\phi_m(x)^\mathsf{T}\Phi_N(x)^{-1}\tilde{\epsilon} = \sum_{n\in[N]}\tilde{\epsilon}_n I^{\text{EZ},n}(\phi_m). \tag{45}$$

Therefore

$$\epsilon^\mathsf{T}\Phi_N(x)^{-1\mathsf{T}}\phi_m(x)\phi_m(x)^\mathsf{T}\Phi_N(x)^{-1}\tilde{\epsilon} = \sum_{n\in[N]}\sum_{n'\in[N]}\epsilon_n\tilde{\epsilon}_{n'}I^{\text{EZ},n}(\phi_m)I^{\text{EZ},n'}(\phi_m), \tag{46}$$

then

$$\mathbb{E}_{\text{DPP}}\epsilon^\mathsf{T}\Phi_N(x)^{-1\mathsf{T}}\phi_m(x)\phi_m(x)^\mathsf{T}\Phi_N(x)^{-1}\tilde{\epsilon} = \sum_{n\in[N]}\sum_{n'\in[N]}\epsilon_n\tilde{\epsilon}_{n'}\mathbb{E}_{\text{DPP}}I^{\text{EZ},n}(\phi_m)I^{\text{EZ},n'}(\phi_m). \tag{47}$$

Now, for $n,n'\in[N]$,

$$\mathbb{E}_{\text{DPP}}I^{\text{EZ},n}(\phi_m) = \int_{\mathcal{X}}\phi_m(x)\phi_n(x)\mathrm{d}\omega(x) = 0, \tag{48}$$

and

$$\mathbb{E}_{\text{DPP}}I^{\text{EZ},n'}(\phi_m) = \int_{\mathcal{X}}\phi_m(x)\phi_{n'}(x)\mathrm{d}\omega(x) = 0. \tag{49}$$

Therefore

$$\mathbb{E}_{\text{DPP}}I^{\text{EZ},n}(\phi_m)I^{\text{EZ},n'}(\phi_m) = \mathbb{Cov}_{\text{DPP}}(I^{\text{EZ},n}(\phi_m), I^{\text{EZ},n'}(\phi_m)). \tag{50}$$

Now, by Proposition 1, we have $\mathbb{Cov}_{\text{DPP}}(I^{\text{EZ},n}(\phi_m), I^{\text{EZ},n'}(\phi_m)) = \delta_{n,n'}$, so that

$$\mathbb{E}_{\text{DPP}}I^{\text{EZ},n}(\phi_m)I^{\text{EZ},n'}(\phi_m) = \delta_{n,n'}, \tag{51}$$

and

$$\mathbb{E}_{\text{DPP}}\epsilon^\mathsf{T}\Phi_N(x)^{-1\mathsf{T}}\phi_m(x)\phi_m(x)^\mathsf{T}\Phi_N(x)^{-1}\tilde{\epsilon} = \sum_{n\in[N]}\sum_{n'\in[N]}\epsilon_n\tilde{\epsilon}_{n'}\mathbb{E}_{\text{DPP}}I^{\text{EZ},n}(\phi_m)I^{\text{EZ},n'}(\phi_m)$$

$$= \sum_{n\in[N]}\sum_{n'\in[N]}\epsilon_n\tilde{\epsilon}_{n'}\delta_{n,n'}$$

$$= \sum_{n\in[N]}\epsilon_n\tilde{\epsilon}_n. \tag{52}$$

Now, for $\boldsymbol{\epsilon} \in \mathbb{R}^N$ and $m \in \mathbb{N}^*$ define $Y_{\boldsymbol{\epsilon},m}$ by

$$Y_{\boldsymbol{\epsilon},m} = \sigma_m \boldsymbol{\epsilon}^{\mathsf{T}} \boldsymbol{\Phi}_N(\boldsymbol{x})^{-1\mathsf{T}} \phi_m(\boldsymbol{x}) \phi_m(\boldsymbol{x})^{\mathsf{T}} \boldsymbol{\Phi}_N(\boldsymbol{x})^{-1} \boldsymbol{\epsilon}. \tag{53}$$

We have

$$\mathbb{E}_{\mathrm{DPP}} Y_{\boldsymbol{\epsilon},m} = \sigma_m \sum_{n \in [N]} \epsilon_n^2, \tag{54}$$

and the $Y_{\boldsymbol{\epsilon},m}$ are non-negative since

$$Y_{\boldsymbol{\epsilon},m} = \sigma_m (\boldsymbol{\epsilon}^{\mathsf{T}} \boldsymbol{\Phi}_N(\boldsymbol{x})^{-1\mathsf{T}} \phi_m(\boldsymbol{x}))^2 \geq 0, \tag{55}$$

moreover,

$$\sum_{m=N+1}^{+\infty} \mathbb{E}_{\mathrm{DPP}} Y_{\boldsymbol{\epsilon},m} < +\infty. \tag{56}$$

Therefore, by Beppo Levi's lemma

$$\mathbb{E}_{\mathrm{DPP}} \sum_{m=N+1}^{+\infty} Y_{\boldsymbol{\epsilon},m} = \sum_{m=N+1}^{+\infty} \mathbb{E}_{\mathrm{DPP}} Y_{\boldsymbol{\epsilon},m}$$

$$= \sum_{n \in [N]} \epsilon_n^2 \sum_{m=N+1}^{+\infty} \sigma_m. \tag{57}$$

Now, in general for $m \in \mathbb{N}^*$ such that $m \geq N + 1$, we have

$$\sigma_m \boldsymbol{\epsilon}^{\mathsf{T}} \boldsymbol{\Phi}_N(\boldsymbol{x})^{-1\mathsf{T}} \phi_m(\boldsymbol{x}) \phi_m(\boldsymbol{x})^{\mathsf{T}} \boldsymbol{\Phi}_N(\boldsymbol{x})^{-1} \tilde{\boldsymbol{\epsilon}} \leq \frac{1}{2}(Y_{\boldsymbol{\epsilon},m} + Y_{\tilde{\boldsymbol{\epsilon}},m}), \tag{58}$$

so that for $M \geq N + 1$, we have

$$\sum_{m=N+1}^{M} \sigma_m \boldsymbol{\epsilon}^{\mathsf{T}} \boldsymbol{\Phi}_N(\boldsymbol{x})^{-1\mathsf{T}} \phi_m(\boldsymbol{x}) \phi_m(\boldsymbol{x})^{\mathsf{T}} \boldsymbol{\Phi}_N(\boldsymbol{x})^{-1} \tilde{\boldsymbol{\epsilon}} \leq \frac{1}{2}\left( \sum_{m=N+1}^{+\infty} Y_{\boldsymbol{\epsilon},m} + \sum_{m=N+1}^{+\infty} Y_{\tilde{\boldsymbol{\epsilon}},m} \right). \tag{59}$$

Therefore, by dominated convergence theorem we conclude that

$$\mathbb{E}_{\mathrm{DPP}} \sum_{m=N+1}^{+\infty} \sigma_m \boldsymbol{\epsilon}^{\mathsf{T}} \boldsymbol{\Phi}_N(\boldsymbol{x})^{-1\mathsf{T}} \phi_m(\boldsymbol{x}) \phi_m(\boldsymbol{x})^{\mathsf{T}} \boldsymbol{\Phi}_N(\boldsymbol{x})^{-1} \tilde{\boldsymbol{\epsilon}} = \sum_{m=N+1}^{+\infty} \sigma_m \sum_{n \in [N]} \epsilon_n \tilde{\epsilon}_n. \tag{60}$$

### A.5  Proof of Theorem 3

Let $g \in \mathcal{E}_N$, and denote $\boldsymbol{\epsilon} = \sum_{n \in [N]} \langle g, \phi_n \rangle_\omega \boldsymbol{e}_n$. Combining Theorem 6 and Theorem 7, we obtain

$$\mathbb{E}_{\mathrm{DPP}} \Big\| \mu_g - \sum_{i \in [N]} \hat{w}_i^{\mathrm{EZ},g} k(x_i, .) \Big\|_{\mathcal{F}}^2 = \sum_{m \geq N+1} \sigma_m \sum_{n \in [N]} \epsilon_n^2. \tag{61}$$

Now let $g \in \mathcal{L}_2(\omega)$, we have

$$\Big\| \mu_g - \sum_{i \in [N]} \hat{w}_i^{\mathrm{EZ},g} k(x_i, .) \Big\|_{\mathcal{F}}^2 = \Big\| \mu_g - \mu_{g_N} + \mu_{g_N} - \sum_{i \in [N]} \hat{w}_i^{\mathrm{EZ},g} k(x_i, .) \Big\|_{\mathcal{F}}^2 \tag{62}$$

$$\leq 2\Big( \| \mu_g - \mu_{g_N} \|_{\mathcal{F}}^2 + \Big\| \mu_{g_N} - \sum_{i \in [N]} \hat{w}_i^{\mathrm{EZ},g} k(x_i, .) \Big\|_{\mathcal{F}}^2 \Big), \tag{63}$$

where $g_N = \sum_{n \in [N]} \langle g, \phi_n \rangle_\omega \phi_n \in \mathcal{E}_N$.

Now, observe that

$$\mu_g^{\gamma,N} = \mu_{g_N}^{\gamma,N}, \tag{64}$$

so that

$$\hat{\boldsymbol{w}}^{\mathrm{EZ},g} = \hat{\boldsymbol{w}}^{\mathrm{EZ},g_N}. \tag{65}$$

Therefore

$$\|\mu_g - \sum_{i \in [N]} \hat{w}_i^{\text{EZ},g} k(x_i, .)\|_{\mathcal{F}}^2 \leq 2\Big(\|\mu_g - \mu_{g_N}\|_{\mathcal{F}}^2 + \|\mu_{g_N} - \sum_{i \in [N]} \hat{w}_i^{\text{EZ},g_N} k(x_i, .)\|_{\mathcal{F}}^2\Big). \quad (66)$$

Now, we have

$$\begin{aligned}
\|\mu_g - \mu_{g_N}\|_{\mathcal{F}}^2 &= \sum_{m \geq N+1} \sigma_m \langle g, \phi_m \rangle_\omega^2 \\
&\leq \sigma_{N+1} \sum_{m \geq N+1} \langle g, \phi_m \rangle_\omega^2 \\
&\leq r_{N+1} \|g\|_\omega^2. \quad (67)
\end{aligned}$$

Moreover, by (26) we have

$$\begin{aligned}
\mathbb{E}_{\text{DPP}} \|\mu_{g_N} - \sum_{i \in [N]} \hat{w}_i^{\text{EZ},g_N} k(x_i, .)\|_{\mathcal{F}}^2 &= \sum_{n \in [N]} \langle g, \phi_n \rangle_\omega^2 r_{N+1} \\
&\leq \|g\|_\omega^2 r_{N+1}. \quad (68)
\end{aligned}$$

Combining (66), (67) and (68), we obtain

$$\|\mu_g - \sum_{i \in [N]} \hat{w}_i^{\text{EZ},g} k(x_i, .)\|_{\mathcal{F}}^2 \leq 4 \|g\|_\omega^2 r_{N+1}. \quad (69)$$

# References

[1] G. Gautier, R. Bardenet, and M. Valko. On two ways to use determinantal point processes for monte carlo integration. In *Advances in Neural Information Processing Systems*, volume 32, 2019.