# OpenReview forum: "An analysis of Ermakov-Zolotukhin quadrature using kernels"
_NeurIPS.cc/2021/Conference — NeurIPS 2021 Poster_

### Official Review · Reviewer_sHED · 2021-07-02

**Rating:** 8
**Confidence:** 4

**Summary:**

This paper studies a classic quadrature rule by Ermakov and Zolotukhin and its connections to kernel quadrature rules with DPPs. The authors show that these quadrature rules can be interpreted as special cases of kernel-based interpolation quadrature rules formulated in the paper. By analyzing the worst-case error in an RKHS for the Ermakov-Zolotukhin quadrature, they provide an upper bound on the worst-case error of kernel quadrature with DPPs that improves upon the upper bound in previous work. Synthetic experiments are provided for validating the obtained theoretical results.

**Limitations And Societal Impact:**

yes

**Main Review:**

[Originality]
To my knowledge, the obtained results are novel and original. This paper makes a nice contribution to the literature on kernel quadrature rules by establishing a connection to an old work by Ermakov and Zolotukhin, which has not been well known in the community.

[Quality]
The paper's quality is high. While I have not checked the proofs in the supplementary materials, the proof sketches in the main body are convincing. The literature review is done adequately.  Experiments are well designed.

[Clarity]
The paper is basically clearly written. While the presentation is adequate for experts, it may be a bit hard to follow for non-experts. A few suggestions:

- The title should be more appealing to a wider audience. For instance, you could include ``DPPs'' in the title.

- You should explain what $g$ and $\omega$ are in a usual setting. I suppose $\omega$ is a probability measure of interest, and usually we set $g(x) = 1$ as a constant function.

- Lines 71-72: ``$e_n$ is the $n$-th element of the canonical basis of $\mathbb{R}^N$'' Can you simply say that this is a $N$-vector with the $n$-th coordinate is $1$ and the rest are $0$?

- Theorem 3.  I think ``In particular" just before Eq. (27) is misleading, because  (27) is not a special case of (26) but needs to proved using (26). Maybe ``moreover'' is appropriate.

- Experiments: Can you explain why the different choices of $g$ lead to different convergence behaviors? These $g$ are all canonical, and I don't know why they result in different behaviors.

- It would be nice if you explain whether it's easy to generate sample points from the DPP with the truncated kernel.

[Significance]
- Theorem 4, which is a direct corollary of Theorem 3, is really nice, in that it provides a sharper error bound for kernel quadrature with DPPs than the previous work by [3]. This previous work's convergence bound is not very tight and does not capture the real empirical performance of kernel quadrature with DPPs. Theorem 4 in the current paper solves this problem, and this is a nice contribution to the literature. Conceptually, this paper's contribution provides theoretical support for the use of DPPs in the construction of nodes for kernel quadrature.


**Time Spent Reviewing:**

16 hours

---

> ### Author Response · Authors · 2021-08-10
> **Response to Reviewer sHED**
>
> We thank you for the substantial amount of time you spent on our article and for your positive assessment. We take into account your suggestions to make the article more accessible to a wider audience.

---

### Official Review · Reviewer_74Tt · 2021-07-13

**Rating:** 4
**Confidence:** 5

**Summary:**

The paper gives a kernel quadrature interpretation for certain interpolatory quadrature rules specified by the property that they integrate exactly all functions in certain finite-dimensional spaces spanned by $L^2$-orthonormal functions. Error estimates in expectation are proved when the integration nodes are sampled from a DPP. The authors call these quadratures Ermakov-Zolotukhin quadratures (EZQ).

**Ethical Concerns:**

-

**Limitations And Societal Impact:**

There is little discussion on limitations of the theoretical results. As I mentioned in the main review, my opinion is that generic results like these tend to be quite difficult to apply to concrete numerical integration problems.

Furthermore, throughout the paper it is assumed that the truncated kernel matrix $\\kappa_N(x)$ is non-singular. While DPP sampling should ensure this, it is not difficult to come up with reasonable nodes for which the matrix is singular. This topic is known as unisolvency in the literature.

**Main Review:**

I am very glad to see that the authors are interested in these topics. Unfortunately, I have to recommend that the paper be rejected. The paper contains two contributions: (1) the kernel quadrature interpretation of what the authors call Ermakov-Zolotukhin quadrature (EZQ) and (2) worst-case error estimates for EZQ (when DPP points are used) in RKHSs. As I will detail below, the first contribution is at most of marginal novelty and demonstrates a marked lack of familiarity with literature on interpolatory quadrature. As to the second contribution, I do not know how interesting someone working with DPPs would find the error estimates. In any case, this contribution is very incremental and the proofs are very straightforward applications of an uncorrelatedness result from [15] (and, if written in a more economically, would probably not require a supplement). I also believe that there is a serious error in the proof of one of the main results. Furthermore, the error estimates are quite generic and difficult to use in most cases if one insists on starting from a given kernel because Mercer decompositions are usually unavailable.

The paper is very well written and, except perhaps for the supplement, a pleasure to read, and would in my opinion suit well for a less prominent venue after some revisions.

ON NOVELTY. The central idea of the paper is that quadrature rules defined by the property that they integrate certain $N$ functions exactly can be interpreted as kernel quadratures for a kernel whose RKHS is $N$-dimensional has appeared many times before, even in machine learning community. As the authors mention, this is done in refs. [21] and [22], but to this list I can add and Fasshauer & McCourt 2012 and Karvonen & Särkkä 2017 (both contain some material on the intermediate case $M > N$) as well as the use of kernels in classical quadrature theory, for which I refer to the review in Section 7.2 in Cools 1997. The extension from polynomials (or related functions) used in these works to orthonormal basis functions used in this paper is rather trivial. The authors may also find material on generalised Gaussian quadratures quite familiar (e.g., Ma et al. 1996).


ON PROOFS AND MATH.
- Equation (63) in the supplement is not true for general $g \\in \\mathcal{L}_2(\\omega)$.

This equation would hold only if we had $\\mu_g(x_i) = \\mu_{g_N}(x_i)$ for every $i \\in [N]$, but this obviously is not the case since the l.h.s is
$$
\\sum_{m > 0} \\sigma_m \\phi_m(x_i) \\langle g, \\phi_m \\rangle_\\omega
$$
while the r.h.s. is
$$
\\sum_{m \\in [N]} \\sigma_m \\phi_m(x_i) \\langle g, \\phi_m \\rangle_\\omega.
$$
This probably complicates or invalidates the proof Equation (27), which is the main result of the paper.
- Lemma 1 in the supplement is unnecessary.

Equation (36) in Proposition 5 can be proved by observing that for $g \\in \\mathcal{E}_N$ the embedding $\\mu_g$ is an element of $\\mathcal{E}_N$. Since the l.h.s. of (36) is the EZQ estimate of the integral of $\\mu_g$ and, by the reproducing property, the r.h.s. equals the integral of $\\mu_g$, it follows from the fact that the EZQ integrates all functions in $\\mathcal{E}_N$ exactly that (36) is true.

- I see little reason to work with $\\epsilon$, $\\epsilon'$, the kernel $k^\\bot$ and bunch of linear algebra as Equation (26) can be proved in a more transparent way as follows:

\\begin{align*}

\\bigg\\lVert \\mu_g - \\sum_{i \\in [N]} \\hat{w}_i^{\\text{EZ},g} k(x_i, \\cdot) \\bigg\\rVert_\\mathcal{F}^2 &= \\bigg\\lVert \\sum_{m > 0} \\sigma_m \\phi_m \\bigg( \\int_\\mathcal{X} \\phi_m(x) g(x) d \\omega(x) - I^{\\text{EZ},g}( \\phi_m ) \\bigg) \\bigg\\rVert_\\mathcal{F}^2 \\\\

&= \\bigg\\lVert \\sum_{m > N} \\sigma_m \\phi_m \\bigg( \\int_\\mathcal{X} \\phi_m(x) g(x) d \\omega(x) - I^{\\text{EZ},g}( \\phi_m ) \\bigg) \\bigg\\rVert_\\mathcal{F}^2 \\\\

&= \\sum_{m,n >N} \\sigma_m \\sigma_n \\langle \\phi_m, \\phi_m \\rangle_\\mathcal{F} \\bigg( \\int_\\mathcal{X} \\phi_m(x) g(x) d \\omega(x) - I^{\\text{EZ},g}( \\phi_m ) \\bigg) \\bigg( \\int_\\mathcal{X} \\phi_n(x) g(x) d \\omega(x) - I^{\\text{EZ},g}( \\phi_n ) \\bigg) \\\\

&= \\sum_{m > N} \\sigma_m I^{\\text{EZ},g}( \\phi_m )^2 .

\\end{align*}

Taking the DPP expectation and, writing $I^{\\text{EZ},g}( \\phi_m ) = \\sum_{n \\in [N]} \\langle g, \\phi_n \\rangle_\\omega I^{\\text{EZ},n}( \\phi_m )$ and using the sum of variance formula and Proposition 1 in the supplement yields Equation (26).


ON THE ROLE OF KERNELS. One of the main points of this paper is that kernel methods help in understanding the Ermakov-Zolotukhin quadrature. I do not completely agree with this and would argue that kernels play merely an incidental role here. The authors take what I would call a kernel-centric approach, selecting a kernel and its Mercer decomposition and constructing the EZQ using the Mercer basis functions. However, one could as well select first an orthonormal basis $(\\phi_m)$ of $\\mathcal{L}_2(\\omega)$ and some non-increasing positive constants $\\sigma_m$. Then the results of the paper could be modified to state that the rate of decay of $\\sigma_m$ controls the rate of convergence the EZQ in the space of functions whose $\\mathcal{L}_2(\\omega)$-coefficients decay sufficiently fast to guarantee that Equation (4) holds. That this space happens to be an RKHS under a summability condition on $\\sigma_m$ can be considered somewhat coincidental. For example, papers by Irrgeher & Leobacher 2015 and Irrgeher et al. 2015 consider different function spaces that arise when the basis functions are Hermite polynomials and $\\sigma_m$ decay with different rates. There is a similar flavour in Zwicknagl & Schaback 2013. Indeed, it is not difficult to derive Equation (26) directly from

$$
\\sup_{ \\lVert f \\rVert_\\mathcal{F} \\leq 1} \bigg \lvert \int f(x) g(x) d \omega(x) - I^{\text{EZ},g}(f) \bigg\rvert^2 = \\sup_{ \\lVert f \\rVert_\\mathcal{F} \\leq 1} \bigg \lvert \sum_{m > N} \langle f, \phi_m \rangle_\omega \bigg( \int \phi_m(x) g(x) d \omega(x) - I^{\text{EZ},g}(\phi_m) \bigg) \bigg\rvert^2,
$$
the fact that $\sqrt{\text{Eq. (4)}} = \\lVert f \\rVert_\\mathcal{F} \\leq 1$ implies that $-\\sqrt{\\sigma_m} \leq \langle f, \phi_m \rangle_\omega \leq \\sqrt{\\sigma_m}$, and Proposition 1 in the supplement.

Now, the kernel-centric approach of the authors is of course useful if one is specifically interested in using a certain kernel and understanding how fast certain quadrature rules converge in its RKHS (e.g., ref [21], Kuo & Wozniakowski 2012 or Kuo et al. 2017; some results of this type are also collected in Chapter 6 of Oettershagen 2017) or in approximating the optimal kernel quadrature weights for computational purposes (e.g., ref [22] or Fasshauer & McCourt 2012). But in general this is complicated by the fact that Mercer expansions (or any RKHS-orthonormal bases) are available only for few kernels. The jump from generic analysis in this paper to something that might be useful in practice tends to be substantial.

ON TERMINOLOGY. The Ermakov-Zolotukhin quadrature is a rather obscure quadrature method whose distinguishing, as I understand it, feature is the use of nodes drawn from a DPP. However, this article (in particular Section 3) effectively renames all quadrature rules whose weights are solved from Equation (17) as EZQs. This terminology is quite non-standard. For example, all Gaussian quadratures would be special cases of the EZQ (select $\\phi_i$ as the orthonormal polynomials for the measure $\\omega$). Quadratures whose weights are defined using an equation like (17), which guarantees that they are exact for functions in an N-dimensional function space, are typically called interpolatory quadrature rules (e.g., Section 6.1 in Cools 1997). Furthermore, I don't quite understand the term "kernel-based interpolation quadrature" introduced in Section 3.1 as each of these quadratures in Equation (20) is simply an optimal kernel quadrature for a kernel whose RKHS is M-dimensional.

SPECIFIC COMMENTS:
- Abstract: "while the weights are defined through a linear system, similarly to the optimal kernel quadrature". This sentence makes it sound like the quadrature weights being defined as the solution of a linear system is a special property of kernel quadrature. However, this is how the weights of any interpolatory quadrature rule (i.e., a quadrature rule which has N points and which is exact for all elements of some N-dimensional function space) can be computed, including Gaussian, Newton-Cotes and Clenshaw-Curtis quadratures. It is just that this linear-algebraic construction is not computationally efficient or stable for classical quadratures and hence rarely used.
- Lines 87-92 & 262-3: These give the impression that using the worst-case error in a function space to assess and compare the accuracy of different quadrature rules is a new or recent innovation.
- Lines 94-95: It may be that Fred Hickernell popularised the use of kernel methods in the QMC community, but the relationship between kernels and numerical integration is much older. Larkin 1970 is the oldest paper that I know of where optimal kernel quadrature appears explicitly.
- The definition of "optimal kernel quadrature" on lines 104-6 is confused: It is first implied that the nodes are fixed ("for a given configuration") but later these appear to be free parameters ("with the nodes and weights that minimize (13)"). The first interpretation seems to be the one that is used in the paper.
- Line 120: While perfectly suitable for numerical integration, the so called $P$-greedy points in [11] and [12] do not encode any knowledge about the integration measure $\\omega$. This may be worth pointing out.
- Lines 169-170: I do not see any reason why this approximation would not be valid also for $g \\notin \\mathcal{E}_N$. In that case the mean embedding vectors defining the EZQ and OKQ weights will of course be different, but for large $N$ the difference will be negligible.
- It would be helpful to provide the closed-form expression that I believe the kernel in Equation (47) has in terms of a Bernoulli polynomial.
- The proof of Theorem 3 is confusingly given partially in the main text and partially in the supplement. For example, currently Equation (60) serves no purpose in the supplement.

MISCELLANEOUS THOUGHTS:
- Taking $\\omega$ as the Gaussian measure and the orthornormal basis functions as Hermite polynomials one can obtain the Mehler kernel. See Irrgeher & Leobacher 2015 and Irrgeher et al. 2015 for related error analysis.
- No $\\mathcal{L}_2(\\omega)$-orthogonality is necessary to construct a quadrature rule with weights like (17) as in general the vector $e_n$ on the r.h.s. consists of integrals of the basis functions. How necessary is orthogonality really? Is it required in DPP-related results?
- One way to derive a worst-case result would be to take supremum in Equation (10). This is of course differs from what the authors do in that DPP-expectation and supremum are taken in different order.

MINOR COMMENTS:
- Lines 54-55: This makes it sound like the eigenfunctions can be taken to be continuous because the eigenvalues are positive and non-increasing, while in reality this follows from the kernel (and hence its RKHS) being continuous.
- Line 58: $\\mathcal{L}(\\omega)$ is missing a subscript.
- Line 125: $\\mathcal{L}(\\omega)$ is missing a subscript.
- Line 133: $\\mathcal{L}(\\omega)$ is missing a subscript.
- Line 186: "In particular" should probably be replaced with "Furthermore" or something similar.
- Subscripts in suprema of Equations (12) and (28) are written differently.
- Line 243: Reference to [5] could be more specific.
- Ref [1]: "Monte carlo"
- Ref [7]: "F. Briol"
- Ref [21]: "hilbert"
- $n'$ is somehow messed up in the supplement.

REFERENCES (note that many of these are only tangentially related to the paper)

- Larkin 1970. Optimal approximation in Hilbert spaces with reproducing kernel functions. Mathematics of Computation.
- Ma, Rokhlin & Wandzura 1996. Generalized Gaussian quadrature rules for systems of arbitrary functions. SIAM Journal on Numerical Analysis.
- Cools 1997. Constructing cubature formulae: the science behind the art. Acta Numerica.
- Fasshauer & McCourt 2012. Stable evaluation of Gaussian radial basisfunction interpolants. SIAM Journal on Scientific Computing.
- Kuo & Wozniakowski 2012. Gauss-Hermite quadratures for functions from Hilbert spaces with Gaussian reproducing kernels. BIT Numerical Mathematics.
- Zwicknagl & Schaback 2013. Interpolation and approximation in Taylor spaces. Journal of Approximation Theory.
- Irrgeher & Leobacher 2015. High-dimensional integration on $\\mathbb{R}^d$, weighted Hermite spaces, and orthogonal transforms. Journal of Complexity.
- Irrgeher, Kritzer, Leobacher & Pillichshammer 2015. Integration in Hermite spaces of analytic functions. Journal of Complexity.
- Karvonen & Särkkä 2017. Classical quadrature rules via Gaussian processes. In 27th IEEE International Workshop on Machine Learning for Signal Processing.
- Kuo, Sloan & Wozniakowski 2017. Multivariate integration for analytic functions with Gaussian kernels. Mathematics of Computation.
- Oettershagen 2017. Construction of Optimal Cubature Algorithms with Applications to Econometrics and Uncertainty Quantification. PhD dissertation.


**Time Spent Reviewing:**

10

---

> ### Author Response · Authors · 2021-08-10
> **Response to Reviewer 74Tt**
>
> We thank you for the detailed review and the enriching discussion of our article. We take into account your suggestions to better situate the contribution of this article compared to existing work on interpolatory quadrature rules/generalized Gaussian quadrature. After double-checking, it seems that there is no error in the passage of the proof that you have mentioned. We have now included this explanation in the supplementary to make the proof clearer.

---

> ### Comment · Reviewer_74Tt · 2021-08-11
> **Response to the authors**
>
> To avoid cluttering discussion on the detailed response, I respond to the authors' comments about my review here.
>
> ----
>
> Gaussian quadrature:
>
> Let me begin by being explicit about the terminology I use. A (generalised) Gaussian quadrature rule refers to a quadrature rule which uses N nodes to integrate 2N functions exactly. An interpolatory rule uses N nodes to integrate (at least) N functions exactly. The sentence "The extension from polynomials (or related functions) used in these works to orthonormal basis functions used in this paper is rather trivial." in my review refers to the interpolatory rules (some of which happen to be Gaussian, but that is not relevant here) which appear in the references earlier in the same paragraph ([21], [22], Fasshauer & McCourt 2012, Karvonen & Särkkä 2017 and Cools 1997). The pointer to generalised Gaussian quadrature merely gives an example of quadrature weights defined as in Equation (17). Finding the nodes of a generalised Gaussian quadrature rule is indeed an almost intractable task.
>
> ----
>
> Error in the proof:
>
> Thank you for the clarification. I see where my mistake in thinking the proof erroneous originated.
>
> ----
>
> Main contribution:
>
> A theoretical result absolutely does not have to have a complicated proof to be relevant or interesting. However, in my opinion publishing such a result in a top venue requires either that (a) the result is of considerable significance or a reasonably long-standing open problem or (b) its simple proof is in some sense non-trivial or insightful (but this alone is often not sufficient). Neither of these conditions is satisfied by this paper:
>
> (a) The main results consist of incremental improvements of the worst-case error bounds in [3] for kernel quadrature with DPPs. This incrementality might not be a problem if the KQ method of [3] was of great significance, but as far as I can see this method is merely one among many recent approaches for choosing the nodes for KQ. Its practical significance is greatly hampered by the need of access the Mercer eigenfunctions. I do acknowledge that in the specific context of DPP-KQ the results are nice (and certainly much nicer than those in [3]); it is publishing in NeurIPS a rather incremental result for a recent method which has not demonstrated its significance that I object to.
>
> (b) At its core the proof consists of nothing more than a straightforward and rather obvious application of a recent equality in DPP literature (if a key auxiliary result were lifted from seeminly unrelated literature this could be exciting). I do not find any part of the proof especially insightful.
>
> To be clear: I would really like to be able to recommend accepting this paper. But there is simply too little going on here that I could imagine seeing the paper in NeurIPS. If this were a different conference, my recommendation would likely be different.
>
> ----
>
> I have updated my score from 3 to 4.

---

> > ### Author Response · Authors · 2021-08-30
> > **Response to Reviewer 74Tt**
> >
> >
> > We thank the reviewer for the additional feedback.
> >
> > It is true that many methods for choosing the nodes of KQ already exist. However, these methods are tailored for specific RKHSs and can hardly be generalized for other RKHSs. Now, DPPs offer a principled methodology to design kernel quadrature nodes, although sampling from the corresponding DPP without the need for explicit Mercer decomposition is still an open problem. This problem is beyond the scope of this article, and we would be glad to tackle it in future work.
> >
> > The proofs of the main results of this article (Theorem 3 and Theorem 4) use a combination of several results proven in the article, and it does not rely entirely on Theorem 1 in [15]. For this reason, we do not agree that the proofs are a straightforward application of existing results in the literature. We mention that the first version of this article (non-published) contained an alternative proof of Theorem 7. This proof is based on the analytical calculation of (42). This analytical proof is heavily technical, and we thought about looking for a simpler proof. Few weeks before the submission, we observed that the proof could be simplified significantly using Theorem 1 in [15], and we opted to submit the article with the simplified proof.

---

### Official Review · Reviewer_LzsW · 2021-07-16

**Rating:** 6
**Confidence:** 2

**Summary:**

The authors improve integration error bounds of Ermakov and Zolotukhin quadrature, and optimal quadrature with DPP sampled nodes for functions from the associated RKHS.

**Ethical Concerns:**

None identified.

**Limitations And Societal Impact:**

This is a theoretical work.

**Main Review:**

Clarity
------------
The paper is written very clearly and I was able to follow it easily.

Novelty
-----------
I am not very familiar with the related works, hence I am not sure if the list of related works is exhaustive that authors provide, but their presentation seems genuine.

Contributions
-----------------
This paper has an easy and quick result improving a bound for optimal quadrature by a factor of N^2 and EZ from a sum of tail of eigenvalues to the largest eigenvalue not in the approximated operator. I was not aware of this being an important open problem since EZ or general DPP sampled nodes is not something I tended to use in practice.

The paper discusses general classes of quadratures where both instances fall, but I do not think this is particularly englighting, and in the context of quadratures one can compartmentalize the rules in many different ways.

While this paper certainly brings contributions in it, it left me with the feeling that the contributions are small, especially in the context that I do not find these EZ or DPP based quadratures very practical. While it is true that finding optimal nodes is indeed NP hard, often this is solved in a greedy way. But this opinion of mine does not diminish the contribution this paper carries with itself. In my opinion it just limits the target audience.

Verdict
--------------
I see no problem in accepting this paper but papers with more practical relevance should be given priority appearing at NeurIPS in opinion, since this paper has, in my opinion, a limited audience.


**Time Spent Reviewing:**

2

---

> ### Author Response · Authors · 2021-08-10
> **Response to Reviewer LzsW**
>
> We thank you for your time reviewing our article. We have discussed the relevance of this article to an ML audience.

---

### Official Review · Reviewer_yKqn · 2021-07-20

**Rating:** 6
**Confidence:** 1

**Summary:**

The authors demonstrate how two kinds of kernel quadrature are related (Ermakov-Zolotukhin quadrature and optimal kernel quadrature) by defining a generalisation that encapsulates both. They then give worst case bounds for Ermakov-Zolotukhin quadrature, which are worse than for the optimal one. However, they then build on this result to tighten the known bounds of optimal kernel quadrature when using a determimental point process to distribute the nodes.

Result is certainly relevant and valuable, though of very niche interest. Still a lot more interesting than most "0.1% better" NN papers though! Paper itself is as clear as a wall of maths can be (and has a good intro), but while the precision is critical I will always object to this mode of presentation: maths written without intuition or examples works only for those who already understand the solution. For everyone else it creates an unnecessary challenge to understanding. But maybe this doesn't matter given the niche nature of the result.

To allow a reader to apply an appropriate weight to this review, it should be known that I have not understood this paper to the standard I would usually obtain. There is too much, I haven't had the time and I remain extremely burned out (or whatever this is; "brain replaced with tired cloud" fits just as well) from the extreme workload of the last year. Saying that, given the shear mathematical denseness within this paper I'm not convinced giving it a proper review in the tight time frame of a conference review is actually a realistic prospect.

**Main Review:**

Maths looks correct, and no errors have jumped out, but I have not checked it to any real depth, so can't be sure. Notation is very verbose at points, even with the occasional simplification - the visual complexity creates an additional barrier to the reader.

Conclusion: "This work shows the importance of the worst-case integration error as a figure of merit when comparing quadrature rules.": Does it? It shows that results can be derived and match experimental results, so I would agree with "useful". But I see no reason why you could conclude, based on the evidence within, that it's anything more than one of many equally valid bounds, none more important than the next.


Grammer etc. issues (all minor):

L11: "interest interesting"
L24: CLTs used without prior definition
L36: RKHS used without prior definition (though words given two lines above).
L55: "We precise"
L67: "integrals that write as"
L99: "this quantity have"
L106: "We precise in"
Figure 1: A separate key, shared by all graphs, could probably be made large enough to be readable. In a paper swarming with equations it seems weird to type "squared error" for the y-axis!
Not sure of the wisdom of having appendix A link to "supplementary.pdf" - I renamed that file to have the name of the paper in (same directory as other reviews) so link was broken.


**Time Spent Reviewing:**

14

---

> ### Author Response · Authors · 2021-08-10
> **Response to Reviewer yKqn**
>
> We thank you for the substantial amount of time and effort you spent on reviewing our article. We take into account your suggestions to make the article more accessible to a wider audience.

---

### Decision · Program_Chairs · 2021-09-27

**Decision:**

Accept (Poster)

**Comment:**

This paper identifies connections between the literatures on kernel quadrature and determinantal point processes and uses these connections to perform theoretical analysis for a quadrature method due to Ermakov and Zolotukhin.  The reviewers agreed on the correctness and the extent of the novelty of the paper (somewhat incremental), but strongly disagreed about its significance, with one reviewer arguing that the Ermakov-Zolotukhin method is not widely used and thus unimportant, and one reviewer championing the paper as being of theoretical interest.  I believe the balance is somewhere in between these extremes - this paper makes a small theoretical contribution, but it is nevertheless one that will interest a subset of the participants at NeurIPS.